# The Benefits of Label-Description Training for Zero-Shot Text Classification

**Lingyu Gao[1], Debanjan Ghosh[2][†], and Kevin Gimpel[1][†]**
[1]Toyota Technological Institute at Chicago
[2]Educational Testing Service
{lygao, kgimpel}@ttic.edu,
dghosh@ets.org

## Abstract

Pretrained language models have improved zero-shot text classification by allowing the transfer of semantic knowledge from the training data in order to classify among specific label sets in downstream tasks. We propose a simple way to further improve zero-shot accuracies with minimal effort. We curate small finetuning datasets intended to describe the labels for a task. Unlike typical finetuning data, which has texts annotated with labels, our data simply describes the labels in language, e.g., using a few related terms, dictionary/encyclopedia entries, and short templates. Across a range of topic and sentiment datasets, our method is more accurate than zero-shot by 17-19% absolute. It is also more robust to choices required for zero-shot classification, such as patterns for prompting the model to classify and mappings from labels to tokens in the model's vocabulary. Furthermore, since our data merely describes the labels but does not use input texts, finetuning on it yields a model that performs strongly on multiple text domains for a given label set, even improving over few-shot out-of-domain classification in multiple settings.

## 1 Introduction

Pretrained language models (PLMs) (Radford et al., 2018; Devlin et al., 2019; Liu et al., 2019; Brown et al., 2020; Raffel et al., 2020) have produced strong results in zero-shot text classification for a range of topic and sentiment tasks, often using a pattern-verbalizer approach (Schick and Schütze, 2021). With this approach, to classify the restaurant review "Overpriced, salty and overrated!", a pattern like "the restaurant is [MASK]" is appended to the review and verbalizers are chosen for each label (e.g., "good" for positive sentiment and "bad" for negative). The text is classified by the pretrained masked language modeling (MLM) head to choose

| Label | Input |
|---|---|
| Business | business
finance
Business is the activity of making one's living or making money by producing or buying and selling products... |
| Sports | sports
racing
An athletic activity requiring skill or physical prowess and often of a competitive nature, as racing, baseball... |

(a) Topic classification

| Label | Input |
|---|---|
| Very Negative | awful
It was *terrible*.
A *horrendous* experience. |
| Very Positive | great
Just *fantastic*.
Overall, it was *outstanding*. |

(b) Sentiment classification

Table 1: A few examples of LABELDESC training data for topic and sentiment classification.

the most probable verbalizer for the [MASK] position.[1] Although effective, the approach is sensitive to the choice of specific pattern/verbalizer pairs, with subtle changes in the pattern, the verbalizer, or both, often having a large impact on performance (van de Kar et al., 2022; Perez et al., 2021).

To alleviate these issues, we propose a simple alternative approach of training on small curated datasets intended to describe the labels for a task. Unlike typical training datasets, which consist of input texts annotated by hand with labels, our data contains only the *descriptions* of the labels. We refer to this data as LABELDESC data and show a few examples for topic and sentiment classification in Table 1. For topic classification, we include a few terms related to the label (e.g., "finance" for "Business", "racing" for "Sports"), a definition of

---

[†] Co-senior authors.

[1]Please refer to Schick and Schütze (2021) for more details on the pattern-verbalizer approach.

the label from dictionary.com (e.g., "An athletic activity . . ." for "Sports"), and a sentence from the opening paragraph of the label's Wikipedia article (e.g., "Business is the activity of . . . " for "Business"). For sentiment classification, we simply use related terms that capture the specific sentiment (e.g., "terrible" for "Very Negative") as well as a few hand-crafted templates (e.g., "It was $t$." where $t$ is a related term).

Next, we finetune pretrained models using the pattern-verbalizer approach on LABELDESC data and evaluate them for text classification. For topic classification, we use patterns and verbalizers from Schick and Schütze (2022) to train on our LA-BELDESC examples by finetuning the model as well as the MLM head (see Section 3 for details). We refer to training on LABELDESC data as LA-BELDESCTRAINING. In experiments, we show that LABELDESCTRAINING consistently improves accuracy (average improvement of 17-19%) over zero-shot classification across multiple topic and sentiment datasets (Table 2). We also show that LA-BELDESCTRAINING can decrease accuracy variance across patterns compared to zero-shot classification (Table 3), thus being less sensitive to the choice of pattern.

We then conduct additional experiments to reveal the value of LABELDESCTRAINING under various circumstances. To study the impact of verbalizer choice, we experiment with uninformative (randomly initialized) and adversarial (intentionally mismatched) verbalizers (Section 4.2.1). While accuracy drops slightly, both settings are still much more accurate than zero-shot classification with its original verbalizers. That is, LABELDESCTRAIN-ING is able to compensate for knowledge-free or even adversarial verbalizer choice. We also compare to finetuning a randomly initialized classifier head without any patterns or verbalizers, again finding accuracy to be higher than zero-shot (Section 4.2.2). Collectively, our results demonstrate that LABELDESCTRAINING leads to strong performance that is less sensitive than zero-shot classification in terms of pattern/verbalizer choice, while also not requiring a pretrained MLM head.

Since LABELDESC data focuses entirely on the labels without seeking to capture the input text distribution, we would hope that it would exhibit stable performance across datasets with the same labels. So, we compare LABELDESCTRAIN-ING to the approach of training on a small super-vised training set from one domain and testing on another (Section 4.2.4). In multiple cases, LA-BELDESCTRAINING actually attains higher accuracy than few-shot supervised learning tested on out-of-domain test sets, even when hundreds of manually labeled training examples are used (albeit from a different input domain).

In summary, this paper shows several benefits of LABELDESCTRAINING. First, once a practitioner identifies a label set of interest for zero-shot classification, it only requires a few minutes to collect the kind of LABELDESC data shown in Table 1, and training on this data improves over zero-shot by 17-19% absolute. Second, LABELDESCTRAIN-ING leads to greater robustness to pattern/verbalizer choice than zero-shot. Third, LABELDESC data are domain independent with regard to the distribution of the inputs; a single LABELDESC training set can be used for any text classification task as long as it contains the same labels. Our experiments show that this independence to input distribution leads to stable accuracy across domains, even attaining higher accuracy than out-of-domain few-shot learning on a few cases.[2]

## 2  Tasks and LABELDESC Datasets

We evaluate on two types of tasks: *topic classification* on AGNews, Yahoo Answers, and DBPedia (Zhang et al., 2015) and *sentiment classification* on the Stanford Sentiment Treebank (SST) (Socher et al., 2013), Yelp Reviews (Zhang et al., 2015), IMDB (Maas et al., 2011), and Amazon Reviews Polarity (Zhang et al., 2015). We consider both binary and 5-way classification for SST and Yelp datasets (denoted as SST-2, SST-5, Yelp-2, and Yelp-5 henceforth) and only binary for IMDB and Amazon (denoted as IMDB and Amz-2 henceforth).[3] Below we describe how we construct LA-BELDESC data for each label set. Dataset statistics as well as all LABELDESC data are in Section A.5 in the Appendix.

**Topic Classification.**    Since labels in topic classification represent general concepts, we use both subjective descriptors of the labels (e.g., related terms) and objective sources of information (e.g., dictionary definition and Wikipedia sentences)

---

[2]Data and code are available at https://github.com/lingyugao/LabelDescTraining.

[3]Our method could be adopted for other tasks like natural language inference (NLI) using templates similar to how we approached sentiment classification. We leave a full exploration to future work.

when selecting LABELDESC data. In particular, we create LABELDESC examples for the label term itself, three related terms, a selected definition from dictionary.com, and the leading sentence from the label's Wikipedia article. As there are typically multiple dictionary.com definitions for our labels, we select a single definition that best aligns with our understanding of the concept underlying the label. We use the leading Wikipedia sentence because it is typically a brief overview/definition of the concept. Most labels in the Yahoo dataset consist of two keywords (e.g., Society & Culture). For these, we use both label terms, definitions for each, and the leading Wikipedia sentences for each.

We did not tune any of these decisions experimentally, so these choices in defining LABELDESC data are almost certainly suboptimal. This suboptimality is especially likely for the "World" label in the AGNews label set. This label reflects international news, but the dictionary definition and Wikipedia article for the term "World" do not capture that sense of the word. Nonetheless, we did not change our procedure for this label because we wanted our results to reflect a real-world implementation of the idea, complete with its limitations for certain labels.

The LABELDESC instances we are using do not contain exhaustive information. We could easily extend the lists of related terms for each topic or use WordNet or other semantic knowledge resources (Zhang et al., 2019). However, one of the goals of this research is to demonstrate how simple it is to choose LABELDESC examples to improve zero-shot classification in very little time.

**Sentiment Classification.** We use a slightly different procedure for sentiment classification. For 5-way sentiment, we use the label verbalizer itself and four synonym terms. In addition, we write four simple templates: "It was $t$.", "A(n) $t$ experience.", "Just $t$.", and "Overall, it was $t$.", where $t$ is the label verbalizer or a synonym. For binary sentiment, we remove the neutral instances, combine the two positive labels ("Very Positive" and "Positive") into one, and combine the two negative labels ("Very Negative" and "Negative") into one. This procedure produces a total of 25 examples per label (5 terms + 5 terms × 4 templates) for 5-way sentiment and 50 examples per label for binary sentiment. Since these LABELDESC instances are domain-independent, we use the same data for both for 5-way sentiment (Yelp-5 and SST-5) and for binary sentiment (Yelp-2, SST-2, IMDB-2, Amz-2).

**Hyperparameter Tuning.** We adhere to the "true" zero-shot setting where hyperparameters cannot be tuned on a development set for the task of interest (Schick and Schütze, 2022). Therefore, we use a separate dataset for hyperparameter tuning - the 20 Newsgroups (20NG, henceforth) (Lang, 1995) - a topic classification dataset with twenty labels. We select only four labels from 20NG for our purposes: *talk.religion.misc*, *rec.autos*, *sci.med*, and *talk.politics.guns*. We chose these four labels because they are sufficiently distinct that we expect tuning to be informative for other real-world classification datasets; many of the other 20NG labels are highly technical or similar to one other, e.g., the pair *comp.sys.ibm.pc.hardware* and *comp.sys.mac.hardware* as well as the pair *comp.os.ms-windows.misc* and *comp.windows.x*. We follow the same strategy as for topic classification above when constructing LABELDESC data for 20NG. The selected hyperparameters are used for both topic and sentiment classifications.

## 3 Experimental Settings

The following settings are used in our experiments. Unless stated otherwise, we use the pretrained RoBERTa-base ($b$) and RoBERTa-large ($l$) models (Liu et al., 2019) for all experiments since RoBERTa is the predominant choice in related zero-shot and dataless research (Schick and Schütze, 2021; van de Kar et al., 2022; Gera et al., 2022). Additionally, for every dataset, we use the entire available *test* sets for evaluation.

**Zero-shot Classification Baseline.** We use the standard "pattern-verbalizer" approach for topic and sentiment classification. The set of verbalizers used can be found in Table 10 in the Appendix. For choosing verbalizers, we follow the choices of Schick and Schütze (2021) for AGNews, Yahoo, Yelp-5, and SST-5. We follow van de Kar et al. (2022) in choosing verbalizers for Yelp-2, SST-2, IMDB, and Amz-2 and we select verbalizers for DBPedia and 20NG ourselves.

Each pattern comprises a prompt including a [MASK] symbol placed before or after the text input, and we aim to predict the masked token. For example, a prompt is added after the input $x$ to frame classification as a question answering task, e.g., "$x$ Question: What is the topic of this newsgroup? Answer: [MASK]." We use RoBERTa-

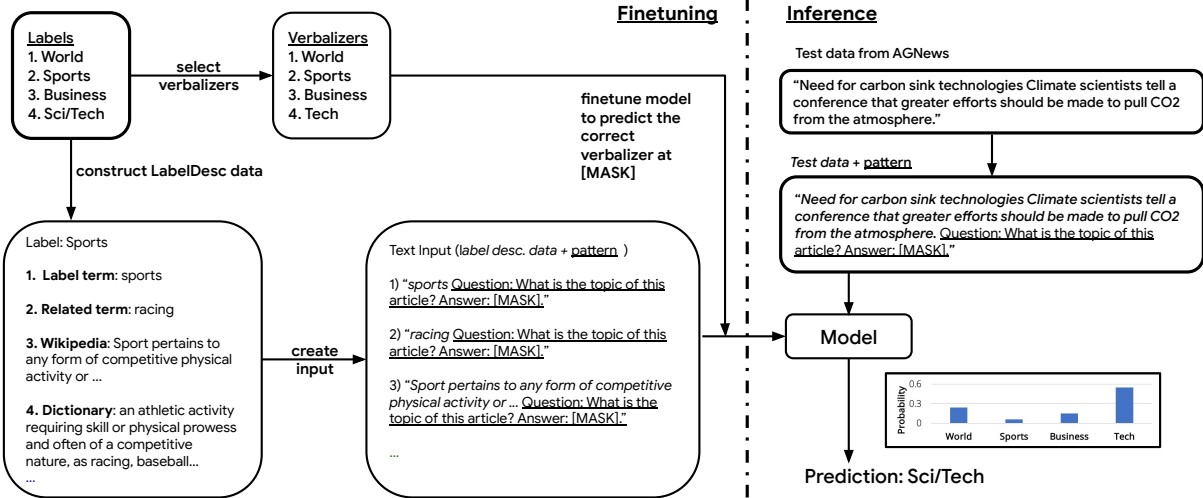

Figure 1: Overview of our proposed method, including the construction of LABELDESC data, the format of the text input, and the target used for both model finetuning and inference during test time. We present text inputs labeled as "Sports" from the topic classification task, and use one of our patterns (see Table 11) here as an illustration. Note that all our LABELDESC datasets are balanced, with each pattern being associated with a unique finetuned model checkpoint.

base/large with its MLM head for zero-shot experiments. Although the model is able to predict any token within its vocabulary, we choose only among the set of verbalizers, which are designed to be semantically coherent with class labels and tokenized into a single token by the model's tokenizer.

For topic classification tasks, we use the PROMPT and Q&A patterns from Schick and Schütze (2022), which amounts to 14 patterns. For AGNews, we use "news/article" in the pattern templates, while for Yahoo we replace this with "question", and for 20NG we use "newsgroup". For the sentiment classification tasks, we create new Q&A patterns such as "$x$ Question: What is the sentiment of this text? Answer: [MASK]." and PROMPT patterns such as "$x$ Sentiment: [MASK]." where $x$ is the input text. There are 14 sentiment patterns in total, presented in the Appendix (Section A.2).

**LABELDESCTRAINING.** We use the same settings as the zero-shot baseline except that we finetune the models on LABELDESC data. We do not use any target task data for tuning or early stopping. Instead, we fix hyperparameter values, including number of training steps, by tuning on 20NG following the process described below.

We used LABELDESC data for the four selected 20NG labels as our training data and the original 20NG data (training and test sets) as our dev set, restricted to the four selected labels shown in Section 2. We preprocessed the data by removing headers,

quotes, and footers. We used a batch size of 1 and tuned over a set of five learning rates ({5e-7, 1e-6, 5e-6, 1e-5, 5e-5}). Models were trained for 3500 training steps, evaluating on the dev set after each epoch, i.e., every 24 training steps since it's the size of LABELDESC dataset for 20NG. Based on tuning accuracies, we chose learning rate 5e-7 and number of training steps 2160 for RoBERTa-base and 1920 for RoBERTa-large. Additionally, we explored variations of parameter freezing, such as freezing certain layers of RoBERTa. The best setting on 20NG was to freeze the lower half of the layers (excluding the embedding layer) during finetuning, so we used this for experiments reported below.[4]

## 4 Results and Analysis

In this section we first present the results that are obtained via LABELDESCTRAINING and then analyze the benefits of LABELDESC data with a range of additional experiments and analysis.

### 4.1 Results

Table 2 compares standard zero-shot classification and LABELDESCTRAINING. LABELDESCTRAINING has higher accuracy across all topic and sentiment classification datasets, outperforming zero-shot by about 17% on average when using

---

[4]Section A.3 in the Appendix provides more details on hyperparameter tuning.

| | | AGNews | Yahoo | DBPedia | Yelp-5 | SST-5 | Yelp-2 | SST-2 | Amz-2 | IMDB | Avg. |
|---|---|---|---|---|---|---|---|---|---|---|---|
| zero-shot | b | 62.7 | 41.5 | 54.6 | 38.0 | 35.6 | 63.6 | 62.6 | 64.0 | 69.9 | 54.7 |
| | l | 68.0 | 47.7 | 63.9 | 38.7 | 35.0 | 70.6 | 63.7 | 67.5 | 74.1 | 58.8 |
| LABELDESCTRAINING | b | 77.4 | 58.8 | 79.5 | 43.6 | 42.0 | 88.3 | 84.5 | 88.6 | 86.9 | 72.2 |
| | l | 79.4 | 60.8 | 86.6 | 51.3 | 49.2 | 94.6 | 91.3 | 94.1 | 92.1 | 77.7 |

Table 2: Test accuracy (%) comparison between zero-shot classification and LABELDESCTRAINING, $b$ = RoBERTa-base, $l$ = RoBERTa-large. For zero-shot, each result is the average over 14 patterns; and for LABELDESCTRAINING, each result is the average over 14 patterns and three random seeds per pattern. The "Avg." column shows the average accuracies across columns.

| | | AGNews | Yahoo | DBPedia | Yelp-5 | SST-5 | Yelp-2 | SST-2 | Amz-2 | IMDB |
|---|---|---|---|---|---|---|---|---|---|---|
| zero-shot | b | 7.4 | 7.0 | 18.9 | 4.3 | 4.3 | 10.7 | 11.0 | 10.3 | 13.2 |
| | l | 7.8 | 8.2 | 9.7 | 7.8 | 7.7 | 15.7 | 14.3 | 13.7 | 17.0 |
| LDT | b | 5.0, 5.1, 5.0 | 1.7, 1.6, 1.6 | 4.5, 4.5, 4.5 | 2.0, 2.1, 2.2 | 1.8, 1.4, 1.5 | 2.1, 2.8, 2.4 | 2.5, 2.3, 1.9 | 1.3, 1.2, 1.4 | 1.8, 2.3, 1.4 |
| | l | 5.3, 6.4, 4.6 | 2.1, 2.0, 2.3 | 3.2, 2.9, 3.2 | 2.4, 2.5, 2.4 | 1.6, 1.2, 1.5 | 1.1, 2.5, 1.4 | 1.2, 2.8, 1.6 | 0.9, 1.9, 0.8 | 1.1, 1.4, 1.2 |

Table 3: Standard deviations of test accuracy (%) across 14 patterns for each test dataset. For LABELDESCTRAINING (LDT in the table), three random seeds were used so we show three standard deviations, one per random seed. All standard deviations over patterns are smaller for LDT than the corresponding values for zero-shot.

RoBERTa-base and 19% with RoBERTa-large. The results demonstrate that we can greatly improve the performance of zero-shot models with just a few training examples that provide a richer characterization of the label but still without requiring any textual inputs from the task datasets.

Table 3 shows that accuracy variances across patterns using LABELDESCTRAINING are much lower than the zero-shot setting, which is known to be unstable (Perez et al., 2021). Finetuning on LABELDESC data not only improves accuracy, but also mitigates sensitivity to pattern selection.

**Comparisons to the State of the Art.** We compare to state-of-the-art (SOTA) results from the literature in Table 4 (we show results using RoBERTa-base to better compare to other methods). For this comparison, we use only a single pattern with LABELDESCTRAINING, since doing so reflects more of a real-world use case than averaging over 14 patterns. We choose a single pattern for each of RoBERTa-base and large by tuning on 20NG as we did for other hyperparameters.[5] We use three random seeds and report average accuracies and standard deviations over seeds.

Chu et al. (2021a) and Chu et al. (2021b) are dataless classification approaches (Chang et al., 2008) that include single-encoder and dual-encoder methods; the latter include the idea of embedding documents and labels and performing classification via semantic retrieval; we report their non-ensemble results in Table 4. Schick and Schütze (2022) use labeled training data (10 or 100 examples, see Table 4) for each task, which differs from the domain-independent LABELDESC examples which are agnostic to the domain of the textual inputs.[6] From van de Kar et al. (2022), we include the highest accuracies.

The results of LABELDESCTRAINING are comparable to other methods across datasets. For sentiment classification, LABELDESCTRAINING performs better than dataless classification (Chu et al., 2021a) by a large margin for all datasets and is competitive with van de Kar et al. (2022) and Schick and Schütze (2021). Our method is better than that of van de Kar et al. on topic datasets (AGNews, Yahoo, and DBPedia) but not sentiment datasets except for SST-2. van de Kar et al. (2022) search for naturally occurring data in large corpora; texts expressing sentiment are well-represented in corpora, while texts for topics in a fixed label set may be rarer. LABELDESCTRAINING trains on balanced data from a fixed label set, leveraging available knowledge resources to inform about topics.

Although van de Kar et al. (2022) do not report 5-way classification results for Yelp or SST, we report results for both datasets (including base and large models) so that future work can compare to our results in this table. We recommend tuning zero-shot and few-shot methods on datasets that

---

[5]Please refer to A.3 and Table 14 in Appendix for details. We use the same setting for Table 5.

[6]We only include results with PROMPT and Q&A patterns (14 patterns for topic and 16 for sentiment) from Schick and Schütze (2022), since those are the pattern types we used for LABELDESCTRAINING.

| | | AGNews | Yahoo | DBPedia | Yelp-5 | Yelp-2 | SST-5 | SST-2 | Amz-2 | IMDB |
|---|---|---|---|---|---|---|---|---|---|---|
| LABELDESCTRAINING | b | 84.6±0.3 | 59.9±0.3 | 82.4±1.2 | 42.0±0.4 | 84.8±0.6 | 44.3±0.1 | 88.2±0.2 | 89.6±0.4 | 83.4±0.4 |
| | l | 85.1±1.0 | 61.2±0.3 | 88.5±0.4 | 52.5±1.2 | 95.3±0.4 | 49.4±1.1 | 91.4±0.8 | 94.5±0.3 | 92.9±0.1 |
| Chu et al. (2021a) | b | 68.8 | 57.8 | 81.9 | - | 67.3 | - | 65.0 | 66.8 | - |
| Chu et al. (2021b) | b | 75.1 | 60.0 | 88.6 | - | - | - | - | - | - |
| Schick and Schütze (2022) | 10 | 79.5±2.2 | 58.4±2.7 | - | 44.3±2.5 | - | - | - | - | - |
| | 100 | 87.5±0.8 | 65.3±1.0 | - | 54.8±1.5 | - | - | - | - | - |
| van de Kar et al. (2022) | b | 79.2 | 56.1 | 80.4 | - | 92.0 | - | 85.6 | 92.0 | 86.7 |

Table 4: Test accuracy (%) comparison to state-of-the-art methods. 10/100 = # labeled examples used.

| | | AGNews | Yahoo | DBPedia | Yelp-5 | Yelp-2 | SST-5 | SST-2 | Amz-2 | IMDB |
|---|---|---|---|---|---|---|---|---|---|---|
| LABELDESCTRAINING | b | 84.3±0.1 | 57.5±0.7 | 82.0±1.5 | 41.6±1.2 | 83.1±0.5 | 45.3±0.6 | 86.7±0.6 | 90.8±0.4 | 83.1±0.6 |
| | l | 85.5±0.6 | 57.5±0.7 | 88.1±0.6 | 53.8±1.9 | 95.4±0.4 | 51.4±1.3 | 90.3±0.7 | 94.2±0.3 | 94.1±0.2 |
| text-davinci-003 (zero-shot) | - | 80.2 | 58.5 | 70.1 | 47.2 | 92.3 | 49.3 | 89.3 | 93.3 | 78.9 |
| text-davinci-003 (ICL) | - | 83.9 | 61.1 | 84.2 | 57.0 | 92.9 | 51.2 | 92.3 | 95.1 | 88.3 |

Table 5: Test accuracy (%) comparison to text-davinci-003 on test set subsets.

are excluded from the final comparison, like 20NG in this paper.

**Comparisons Involving GPT-3.5.** Our method not only works for MLM-style models like RoBERTa, but also for autoregressive models. In Table 5, we show zero-shot and in-context learning (ICL), where we use the entire LABELDESC data for the task as ICL demonstrations, with text-davinci-003 (GPT-3.5; OpenAI, 2022). Due to our restricted budget, we decided to use only 1,000 test instances for each test dataset in GPT-3.5 experiments, while ensuring that the label distribution remains consistent with that of the full test dataset. It is well known that ICL is sensitive to a variety of design choices, including the order of the demonstrations (Fei et al., 2023; Lu et al., 2022). For ICL demonstrations, we included all LABELDESC data for a task to make predictions for each test instance. To avoid the "recency bias" (i.e., the tendency to predict labels that occur towards the end of the prompt; Zhao et al., 2021a), we randomly shuffle the order of demonstrations. We left other parameters untouched. GPT-3.5 with ICL using LABELDESC data outperforms zero-shot GPT-3.5 on all datasets, showing the value of LABELDESC data even if in-domain inputs are unavailable. In comparison to GPT-3.5 flavors, LABELDESCTRAINING (RoBERTa-large) performs better on AGNews, DBPedia, Yelp-2, SST-5, and IMDB, and is competitive across other datasets.

## 4.2 Analysis and Discussion

One of the primary requirements of the zero-shot approach is the availability of pattern-verbalizer pairs (Schick and Schütze, 2021, 2022). Here, we study several variations of LABELDESCTRAINING to investigate whether we can simplify or remove components of these pattern-verbalizer pairs. We first experiment with changing verbalizers to gauge the impact of verbalizer choice for LABELDESCTRAINING (Section 4.2.1). Next, we conduct classification experiments that do not use patterns or verbalizers at all (Section 4.2.2).

Furthermore, we include one more baseline, i.e., the model finetuned on the 20NG LABELDESC data and patterns to analyze the generalizability (Section 4.2.3). We also report additional experiments in which we measure the multi-domain robustness of LABELDESCTRAINING compared to a standard procedure of training on one domain and testing on an out-of-domain test set (Section 4.2.4). Finally, we take a closer look at label-wise performance to better understand how LABELDESCTRAINING outperforms zero-shot classification (Section 4.2.5).

### 4.2.1 Impact of Verbalizers

In this section we report experiments with LABELDESCTRAINING without meaningful verbalizers and even with adversarially chosen verbalizers. We explore two different verbalizer settings:

- RANDOM: We add $c$ new words, i.e., RANDOM1, RANDOM2, ..., RANDOM$c$, where $c$ is the number of dataset labels, to the model's vocabulary and randomly initialize their embeddings. This setting prevents the use of any prior knowledge in the verbalizer embeddings.

- MISMATCHED: We shuffle the original mapping

| | | AGNews | Yahoo | DBPedia | Yelp-5 | SST-5 | Yelp-2 | SST-2 | Amz-2 | IMDB | Avg. |
|---|---|---|---|---|---|---|---|---|---|---|---|
| zero-shot | $b$ | 62.7±7.4 | 41.5±7.0 | 54.6±18.9 | 38.0±4.3 | 35.6±4.3 | 63.6±10.7 | 62.6±11.0 | 64.0±10.3 | 69.9±13.2 | 54.7±9.7 |
| | $l$ | 68.0±7.8 | 47.7±8.2 | 63.9±9.7 | 38.7±7.8 | 35.0±7.7 | 70.6±15.7 | 63.7±14.3 | 67.5±13.7 | 74.1±17.0 | 58.8±11.3 |
| LDT$_{20NG}$ | $b$ | 61.8±7.0 | 49.4±5.2 | 72.9±7.8 | 34.6±4.6 | 36.5±3.7 | 67.7±10.3 | 63.4±9.7 | 67.2±9.6 | 72.5±10.5 | 58.4±7.6 |
| | $l$ | 72.4±6.8 | 54.4±4.3 | 71.9±10.8 | 36.3±5.7 | 36.6±7.1 | 63.4±13.0 | 56.9±8.7 | 60.9±10.2 | 67.5±15.2 | 57.8±9.1 |
| LDT | $b$ | 77.4±4.9 | 58.8±1.6 | 79.5±4.4 | 43.6±2.1 | 42.0±1.6 | 88.3±2.5 | 84.5±2.2 | 88.6±1.4 | 86.9±1.8 | 72.2±2.5 |
| | $l$ | 79.4±5.0 | 60.8±2.1 | 86.6±3.0 | 51.3±2.4 | 49.2±1.6 | 94.6±1.8 | 91.3±2.0 | 94.1±1.3 | 92.1±1.2 | 77.7±2.3 |
| MLM$_r$ | $b$ | 77.3±4.0 | 54.3±3.9 | 81.3±7.3 | 38.1±3.8 | 37.0±3.2 | 78.4±10.0 | 73.3±7.9 | 80.0±9.9 | 73.8±9.6 | 65.9±6.6 |
| | $l$ | 75.2±5.0 | 58.0±3.0 | 85.4±13.0 | 46.4±3.3 | 43.4±2.9 | 90.8±7.6 | 84.1±6.8 | 90.2±7.1 | 87.4±6.2 | 73.4±6.1 |
| MLM$_m$ | $b$ | 73.1±5.6 | 50.1±5.4 | 72.6±8.1 | 36.8±2.8 | 35.8±2.5 | 80.1±7.2 | 75.8±5.0 | 81.8±6.8 | 76.7±6.0 | 64.8±5.5 |
| | $l$ | 66.4±8.6 | 44.5±4.9 | 73.1±7.3 | 41.9±4.0 | 38.7±4.2 | 83.6±6.5 | 78.1±6.0 | 85.0±6.0 | 77.7±6.9 | 65.4±6.0 |
| classifier | $b$ | 72.5±5.5 | 57.1±0.7 | 87.7±2.6 | 40.3±1.3 | 39.4±2.5 | 86.9±2.9 | 79.7±1.1 | 89.1±0.9 | 80.6±3.6 | 70.4±2.3 |
| | $l$ | 77.8±1.5 | 50.9±7.3 | 78.2±1.0 | 42.4±1.6 | 35.3±9.2 | 93.3±0.9 | 86.6±1.4 | 93.7±0.5 | 85.7±2.0 | 71.5±2.8 |

Table 6: Test accuracies (%) for several variations of LABELDESCTRAINING. The standard deviations are computed over 14 patterns for zero-shot; 3 random seeds for the classifier (no patterns); and both 14 patterns and 3 random seeds for LABELDESCTRAINING on 20NG, LABELDESCTRAINING, RANDOM, and MISMATCHED (LDT$_{20NG}$, LDT, MLM$_r$, and MLM$_m$ in Table).

of labels to verbalizers, ensuring that each verbalizer maps to a different label than in the original LABELDESCTRAINING setting. Since we are still finetuning the embeddings, finetuning can help the model recover from this mismatched initialization.

The results are shown in Table 6. Since we still use the MLM head for these results, we refer to them as "MLM, RANDOM" and "MLM, MISMATCHED". While LABELDESCTRAINING performs better than RANDOM, and RANDOM is better than MISMATCHED, both are better than zero-shot on average. These results suggest that LABELDESC data can partially compensate when the quality of the verbalizers is unknown or poor, at least to improve over zero-shot.

### 4.2.2 Classifiers Without Patterns or Verbalizers

Since finetuning on LABELDESC data outperforms zero-shot results with RANDOM verbalizers, we also evaluate its performance without patterns, i.e., using a standard randomly initialized softmax classifier. The input is the original text without any patterns and we use a two-layer classification head on top of the [CLS] token representation of the pretrained models.

The bottom two rows of Table 6 show the results. The classifiers are close to that of the MLM/RANDOM setting and still much higher than zero-shot on average, suggesting that it is not necessary to use patterns, verbalizers, or even the pretrained MLM head in order to outperform zero-shot classifiers. If it is difficult to select verbalizers or design patterns for a particular classification task,

using a classifier that has been finetuned on a small LABELDESC dataset may serve as a strong alternative to the pattern-verbalizer approach.

### 4.2.3 Cross-Task Generalizability

We report results on the model finetuned on the 20NG LABELDESC data and patterns, i.e., LABELDESCTRAINING on 20NG (LDT$_{20NG}$), in Table 6. While the patterns for the reported datasets are different from those used for 20NG, especially for sentiment datasets, they have similar structures (see Section A.2). For RoBERTa-base, LDT$_{20NG}$ often outperforms zero-shot results, except for AGNews and Yelp-5. However, for RoBERTa-large, while LDT$_{20NG}$ outperforms the zero-shot results on all topic classification datasets, it's worse on sentiment classification except for SST-5.

### 4.2.4 Multi-Domain Evaluation

Since LABELDESC examples are domain-independent, they can be used for multiple datasets that have the *same* labels. To assess the multi-domain performance of LABELDESCTRAINING, we compare it to supervised few-shot learning in which a model is trained on data from one domain and then evaluated on a different domain with the same label set (i.e., training on SST-5 and evaluating on Yelp-5). To create multi-domain test sets for a single topic label set, we keep AGNews as it is and create a new subsampled version of Yahoo as follows: (1) "Politics & Government" and "Society & Culture" texts are assigned the label "World", (2) "Sports" texts are labeled "Sports", (3) "Business & Finance" texts are labeled "Business", and (4) "Science & Mathematics" and "Computers

& Internet" texts are labeled "Sci/Tech". Other Yahoo texts are removed. We refer to this new version of the Yahoo dataset as Yahoo_AG. For sentiment classification, we choose two dataset pairs that share label sets, i.e., SST-5 and Yelp-5.

We do not change anything about the LABELDESCTRAINING configuration for these experiments. We simply evaluate the same model on multiple test sets, reporting average accuracies over patterns.

For few-shot setup, we create datasets with 10, 100, and 500 training examples per label. For *in-domain* experiments, train, dev, and test sets are drawn from the same domain/dataset, whereas for *out-of-domain* experiments, train and dev sets are drawn from one domain and the test set is drawn from another domain. We tune learning rates over the same ranges as mentioned earlier and use batch sizes 1, 2, and 4 for 10, 100, and 500 examples per label, respectively. We train for 15 epochs and select the checkpoint from the best epoch selected by the dev set.

The results using RoBERTa-large are shown in Figure 2. For brevity, we only show a subset of results.[7] As we would expect, testing on out-of-domain data leads to accuracy drops but adding more out-of-domain training data reduces this gap. LABELDESCTRAINING, shown as an orange dotted line, outperforms supervised few-shot learning in some cases, such as training on AGNews and testing on Yahoo_AG, even with 500 examples per label (upper-right plot in Figure 2). We see the same trend when the supervised model is trained on Yelp-5 and tested on SST-5 (lower-right plot in Figure 2). In 3 out of 4 cases, LABELDESCTRAINING outperforms supervised few-shot out-of-domain learning with 10 examples per label, outperforming 100 in 2 out of 4 cases.

#### 4.2.5 Label-wise Investigation

To better understand why LABELDESCTRAINING outperforms zero-shot, we report label-specific F1 scores in Tables 8 and 9. For AGNews, the zero-shot classifiers have low F1 scores for the World label, probably because the verbalizer "World" is much less coherent and less representative of the actual label than others like "Sports." LABELDESCTRAINING improves F1 on the World label by roughly 20 points, while the improvement for Sports is only about 4 points. Likewise, the F1 scores for "Very Negative", "Very Positive", and

---
[7]Section A.4 in the Appendix shows additional results.

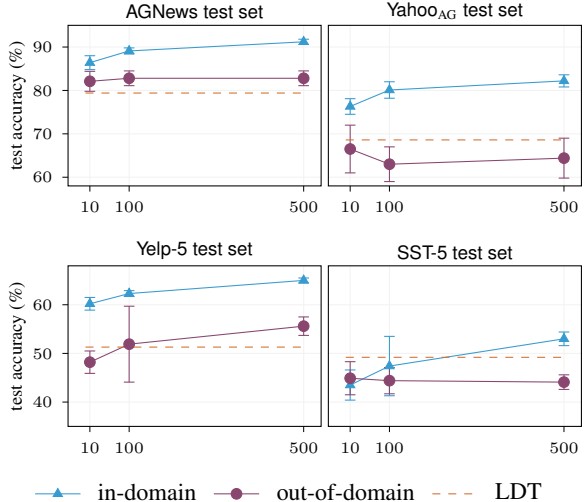

Figure 2: Domain transfer results, where the X-axis shows the number of training examples per label.

"Neutral" are very low for the zero-shot models on SST-5, indicating that those labels are being largely ignored. Again, LABELDESCTRAINING shows large improvements in F1 for some of these labels, especially "Very Positive". These trends are likely due in part to the differences verbalizer probabilities, e.g., "good" and "bad" occur more frequently than "great" and "terrible". The LABELDESC data is balanced, which helps to mitigate the ignoring of labels, even though the task test sets are not all balanced. Table 7 shows examples that are incorrectly classified by zero-shot models but are correctly classified by the LABELDESCTRAINING models.

## 5 Related Work

One common approach in zero-shot text classification is to transfer knowledge from seen labels (Dauphin et al., 2014), which requires observed labels and a notion of label similarity. Some sources of semantic knowledge used for this purpose include multiple modalities (Lampert et al., 2009), label relationships in knowledge graphs (Wang et al., 2018), and word representations (Song and Roth, 2014; Fei et al., 2022).

There are several other approaches to zero-shot classification. To classify documents, Chang et al. (2008) used knowledge-based text representations derived from Wikipedia, and Barak et al. (2009) used both Wikipedia and WordNet. Zhang et al. (2019) combined label descriptions with a label hierarchy and word-to-label paths in ConceptNet, with data augmentation strategies. Yin et al. (2019) used a textual entailment approach with label defi-

| text ([headline][text body] for AGNews) | zero-shot | LABELDESCTRAINING |
|---|---|---|
| [Homeless families total 100,000][The figure for homeless families in England has topped 100,000 for the first time.] | Business | World |
| [Shifting signs in North Korea][Kim Jong Il dials back his personality cult as protest activities pick up.] | Sports | World |
| [GM, Daimler Go Green][Team-up will help the companies compete and fill gaps in both firms' portfolios.] | Sci/Tech | Business |
| (U)nrelentingly stupid. | Positive | Very Negative |
| Still, I'm not quite sure what the point is... | Positive | Negative |
| This 72-minute film does have some exciting scenes, but it's a tad slow. | Positive | Neutral |

Table 7: AGNews/SST-5 data that are correctly classified with LABELDESCTRAINING but not in zero-shot settings.

| | zero-shot | LABELDESCTRAINING |
|---|---|---|
| World | $61.5_{\pm 15.1}$ | $81.0_{\pm 4.3}$ |
| Business | $63.6_{\pm 7.1}$ | $74.9_{\pm 4.7}$ |
| Sports | $88.2_{\pm 3.9}$ | $92.7_{\pm 4.5}$ |
| Sci/Tech | $55.0_{\pm 11.4}$ | $67.8_{\pm 9.3}$ |

Table 8: AGNews label-wise F1 (RoBERTa-large).

| | zero-shot | LABELDESCTRAINING |
|---|---|---|
| Very Negative | $11.2_{\pm 14.9}$ | $25.8_{\pm 5.7}$ |
| Negative | $37.6_{\pm 21.2}$ | $62.5_{\pm 2.0}$ |
| Neutral | $1.2_{\pm 2.9}$ | $10.8_{\pm 5.5}$ |
| Positive | $46.0_{\pm 5.8}$ | $48.2_{\pm 4.9}$ |
| Very Positive | $12.1_{\pm 15.0}$ | $58.0_{\pm 4.0}$ |

Table 9: SST-5 label-wise F1 (RoBERTa-large).

nitions from WordNet. Another approach that has gained popularity is self-training given label names and mining an unlabeled dataset (Meng et al., 2020; Gera et al., 2022). van de Kar et al. (2022) extend the mining-based approach by selecting unsupervised examples (via patterns) for training. Basile et al. (2022) select label descriptions by aggregation. Meng et al. (2022) use language models to generate new training examples. On the contrary, we train on a small set of domain-independent label descriptions. Our setup is influenced by Schick and Schütze (2021, 2022), although, instead of finetuning on training examples, we only use our LABELDESC data.

Autoregressive language models have also been used for zero-shot text classification; we report zero-shot and ICL results with LABELDESC data using GPT-3.5 (OpenAI, 2022). Zhao et al. (2021b) found it beneficial to "calibrate" such models for this setting; this idea is not immediately applicable here due to our use of encoder-only models like RoBERTa. Calibration could be extended to encoder-only models, which we plan to explore in future work. Our work is closely related to data-

less classification (Chang et al., 2008) which involves building classifiers by designing or learning a generic function that scores the compatibility of a document and label defined in natural language. We compared empirically to the dataless classification approaches of Chu et al. (2021a) and Chu et al. (2021b) who used pretrained models, naturally annotated data like that from Wikipedia categories, and unsupervised clustering techniques. There is a wealth of prior work in semi-supervised text classification (Nigam et al., 2000; Xie et al., 2020; Howard and Ruder, 2018). There is also related work on generating label names (Schick et al., 2020) or label descriptions (Chai et al., 2020; Sun et al., 2019) but for supervised text classification.

## 6 Conclusions

We presented LABELDESCTRAINING, a method for improving the accuracy of zero-shot classification by using small, curated datasets that simply describe the labels for a task in natural language. Our method is 17-19% more accurate than zero-shot on average across a range of datasets. LABELDESCTRAINING is also more robust to the choices required for zero-shot classification, such as patterns and verbalizers. Furthermore, LABELDESC data is domain agnostic and therefore can used for any text classification task as long as it contains the same set of labels. LABELDESCTRAINING can even outperform a supervised approach that uses training data from a different domain. One future direction would be to apply the idea to structured prediction, NLI, and natural language generation tasks. Another would be to investigate ways to reduce the dependence of pretrained models on patterns and verbalizers, such as directly calibrating the marginal probabilities of verbalizers with the goal of minimizing biases of pretrained models.

## 7  Limitations

We focus on a simple approach of curating small finetuning datasets that describe the labels for text classification tasks. Although this is beneficial when the task is specific, especially when the data is difficult to obtain, the data curation process is intrinsically intuitive and relies on the practitioner's understanding of the labels and usage situation. Moreover, since a pretrained model is necessary for this approach, a few curated examples may mitigate, but cannot detect or eliminate, potential biases of the pretrained model. If the labels of a certain classification task are dissimilar from the examples the model was trained on, and the model lacks the knowledge to differentiate among them, it may lead to unsatisfying performance even after finetuning on a few examples of label descriptions.

## 8  Ethics Statement

We use pretrained models for text classification, and curate data with the assistance of data sources such as Wikipedia and dictionary definitions. The large pretrained models are trained on a massive amount of data and have been shown to have issues with bias; however, this is a common challenge when working with pretrained models and would benefit from advances made by the community on this front. While both dictionary.com definitions and Wikipedia are aimed at providing accurate and neutral information for a word/concept, they can be affected by the biases and limitations of their editors, especially for Wikipedia, which is an open-source encyclopedia. Our method is not reliant on specific dictionaries or encyclopedias; others could be used. We chose these resources for simplicity as they are highly accessible and widely used. Since our LABELDESC data is very small in size, we manually examined the data as we selected it for any potential biases or other issues. Finally, we use standard topic and sentiment datasets for evaluation, which are used in a great deal of prior work.

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

# A Appendix

## A.1 Verbalizers

| Dataset | Verbalizers |
|---|---|
| 20NG | talk.religion.misc $\mapsto$ religion, rec.autos $\mapsto$ automobile, sci.med $\mapsto$ medicine, talk.politics.guns $\mapsto$ gun |
| AGNews | World $\mapsto$ World, Sports $\mapsto$ Sports, Business $\mapsto$ Business, Sci/Tech $\mapsto$ Tech |
| Yahoo | Society & Culture $\mapsto$ Society, Science & Mathematics $\mapsto$ Science, Health $\mapsto$ Health, Education & Reference $\mapsto$ Education, Computers & Internet $\mapsto$ Computer, Sports $\mapsto$ Sports, Business & Finance $\mapsto$ Business, Entertainment & Music $\mapsto$ Entertainment, Family & Relationships $\mapsto$ Relationship, Politics & Government $\mapsto$ Politics |
| DBPedia | Company $\mapsto$ company, Educational institution $\mapsto$ school, Artist $\mapsto$ artist, Athlete $\mapsto$ sports, Office holder $\mapsto$ politics, Mean of transportation $\mapsto$ transportation, Building $\mapsto$ building, Natural place $\mapsto$ natural, Village $\mapsto$ village, Animal $\mapsto$ animal, Plant $\mapsto$ plant, Album $\mapsto$ album, Film $\mapsto$ film, Written work $\mapsto$ book |
| Yelp-5 SST-5 | Very Negative $\mapsto$ terrible, Negative $\mapsto$ bad, Neutral $\mapsto$ okay, Positive $\mapsto$ good, Very Positive $\mapsto$ great |
| Yelp-2 SST-2 IMDB Amz-2 | Negative $\mapsto$ awful, Positive $\mapsto$ great |

Table 10: Verbalizers selected for each dataset.

## A.2 Patterns for MLM

### A.2.1 Topic Classification

We use the patterns shown in Table 11 for AGNews and DBPedia, and replace "news/article" by "question" for Yahoo Question, which follows Schick and Schütze (2022)'s practice. We use "newsgroup" instead of "question" for 20NG.

### A.2.2 Sentiment Classification

Our sentiment classification datasets (Yelp-2/5, SST-2/5, Amz-2, and IMDB) share the same patterns listed in Table 12.

## A.3 Hyperparameters and Best Pattern

We selected training batch size as 1 for our experiments on LABELDESC data. After fine-tuning on 20NG, the hyperparameters are selected as shown in Table 13. With the selected hyperparameters, we further examine the dev accuracy on 20NG for all prompt patterns and select the tuned pattern that

| type | id | patterns |
|---|---|---|
| Q&A | 1 | $x$ Question: What is the topic of this article? Answer: [MASK]. |
| | 2 | $x$ Question: What is the category of this article? Answer: [MASK]. |
| | 3 | $x$ Question: What is the topic of this article? Answer: [MASK] |
| | 4 | $x$ Question: What is the category of this article? Answer: [MASK] |
| PROMPT | 1 | $x$ Category: [MASK]. |
| | 2 | $x$ Class: [MASK]. |
| | 3 | $x$ Topic: [MASK]. |
| | 4 | $x$ Theme: [MASK]. |
| | 5 | $x$ Category: [MASK] |
| | 6 | $x$ Class: [MASK] |
| | 7 | $x$ Topic: [MASK] |
| | 8 | $x$ Theme: [MASK] |
| | 9 | [MASK] News: $x$ |
| | 10 | [MASK] NEWS: $x$ |

Table 11: Patterns for AGNews, where $x$ refers to the given text.

has the highest dev accuracy. The tuned patterns are listed in Table 14.

To our knowledge, this method works well when we adapt to other datasets. However, we also observe that there are fluctuations in the dev accuracy curve for 20NG during training, and we select the training steps in the middle of the flatter part of curves rather than the peak point for robustness. We suggest changing training steps or increasing batch size if this method doesn't work well.

The tuned pattern is not necessarily the best pattern after adapting to other datasets, sometimes even a little lower than the average results over all 14 patterns.

## A.4 Domain Transfer

All results on RoBERTa-base/large are shown in Figure 3.

## A.5 LABELDESC Data

The statistics of LABELDESC data are shown in Table 15. We use the same set of LABELDESC data for AGNews and Yahoo_AG, Yelp-5 and SST-5, Yelp-2 and SST-2, respectively. The data is listed in Table 16 - Table 21. Each term/sentence that is separated by "|" in tables is an independent LABELDESC example during training. For brevity, we list all hand-crafted templates instead of listing all data for sentiment classification.

| type | id | patterns |
|------|-----|----------|
| Q&A | 1 | $x$ Question: What is the sentiment of this text? Answer: [MASK]. |
| | 2 | $x$ Question: What is the writer's opinion in this text? Answer: [MASK]. |
| | 3 | $x$ Question: What is the sentiment of this text? Answer: [MASK] |
| | 4 | $x$ Question: What is the writer's opinion in this text? Answer: [MASK] |
| PROMPT | 1 | $x$ Opinion: [MASK]. |
| | 2 | $x$ Feeling: [MASK]. |
| | 3 | $x$ Sentiment: [MASK]. |
| | 4 | $x$ Summary: [MASK]. |
| | 5 | $x$ Opinion: [MASK] |
| | 6 | $x$ Feeling: [MASK] |
| | 7 | $x$ Sentiment: [MASK] |
| | 8 | $x$ Summary: [MASK] |
| | 9 | [MASK] Sentiment: $x$ |
| | 10 | [MASK] SENTIMENT: $x$ |

Table 12: Patterns for sentiment classification, where $x$ refers to the given text.

| | | | lr | steps |
|---|---|---|-----|-------|
| MLM | LDT | base | 5e-7 | 2160 |
| | | large | 5e-7 | 1920 |
| | MISMATCHED | base | 5e-5 | 2160 |
| | | large | 5e-6 | 3000 |
| | RANDOM | base | 5e-5 | 2160 |
| | | large | 5e-6 | 3240 |
| classifier | | base | 1e-5 | 1920 |
| | | large | 1e-6 | 2280 |

Table 13: Hyperparameters (learning rate, training steps) selected by tuning on 20NG with RoBERTa.

## A.6 Dataset Preprocessing

For 20NG, we remove headers, quotes, and footers. For AGNews, we concatenate the headlines and the text body of the news articles. For Yahoo dataset, we concatenate the title, the question, and the top answer to it. And for IMDB and Amazon Reviews Polarity datasets, we concatenate the title and the content.

## A.7 Label-wise Metrics

We list label-wise precision, recall, and F1 scores for part of our datasets in Table 22 - 29.

| | | | pattern | id |
|---|---|---|---------|-----|
| MLM | LDT | base | prompt | 9 |
| | | large | prompt | 7 |
| | MISMATCHED | base | qa | 3 |
| | | large | qa | 1 |
| | RANDOM | base | qa | 3 |
| | | large | prompt | 6 |

Table 14: Tuned pattern and pattern id for each model.

| dataset | #label | LD | dev | test |
|---------|--------|-----|------|------|
| 20NG | 4 | 24 | 3389 | - |
| AGNews | 4 | 24 | 2,000 | 7,600 |
| Yahoo$_{AG}$ | | | 3,000 | 36,000 |
| Yahoo | 10 | 60 | - | 60,000 |
| DBPedia | 14 | 84 | - | 70,000 |
| Yelp-5 | 5 | 125 | 2,500 | 50,000 |
| SST-5 | | | 1,101 | 2,210 |
| Yelp-2 | 2 | 100 | 2,000 | 38,000 |
| SST-2 | | | 872 | 1,821 |
| Amz-2 | | | - | 400,000 |
| IMDB | | | - | 25,000 |

Table 15: Statistics of datasets we used, with '#' denoting the number of labels, LD refers to LABELDESC data.

| Label | Type | Training Data |
|-------|------|---------------|
| talk. religion. misc | terms | religion \| Christian \| Buddhist \| Jewish |
| | Wiki. | Religion is usually defined as a social-cultural system of designated behaviors and practices, morals, beliefs, worldviews, texts, sanctified places, prophecies, ethics, or organizations, that generally relates humanity to super-natural, transcendental, and spiritual elements; however, there is no scholarly consensus over what precisely constitutes a religion. |
| | dict. | a set of beliefs concerning the cause, nature, and purpose of the universe, especially when considered as the creation of a superhuman agency or agencies, often involving devotional and ritual observances, and often containing a moral code governing the conduct of human affairs. |
| rec.autos | terms | automobile \| truck \| car \| vehicle |
| | Wiki. | A car (or automobile) is a wheeled motor vehicle that is used for transportation. |
| | dict. | a passenger vehicle designed for operation on ordinary roads and typically having four wheels and a gasoline or diesel internal-combustion engine. |
| sci.med | terms | medicine \| hospital \| symptom \| cure |
| | Wiki. | Medicine is the science and practice of caring for a patient, managing the diagnosis, prognosis, prevention, treatment, palliation of their injury or disease, and promoting their health. |
| | dict. | any substance or substances used in treating disease or illness; medicament; remedy. |
| talk. politics. guns | terms | gun \| firearm \| weapon \| handgun |
| | Wiki. | A gun is a ranged weapon designed to use a shooting tube (gun barrel) to launch projectiles. |
| | dict. | a weapon consisting of a metal tube, with mechanical attachments, from which projectiles are shot by the force of an explosive; a piece of ordnance. |

Table 16: LABELDESC data for 20NG. "Wiki." and "dict." refers to the data source, i.e., Wikipedia or dictionary definition.

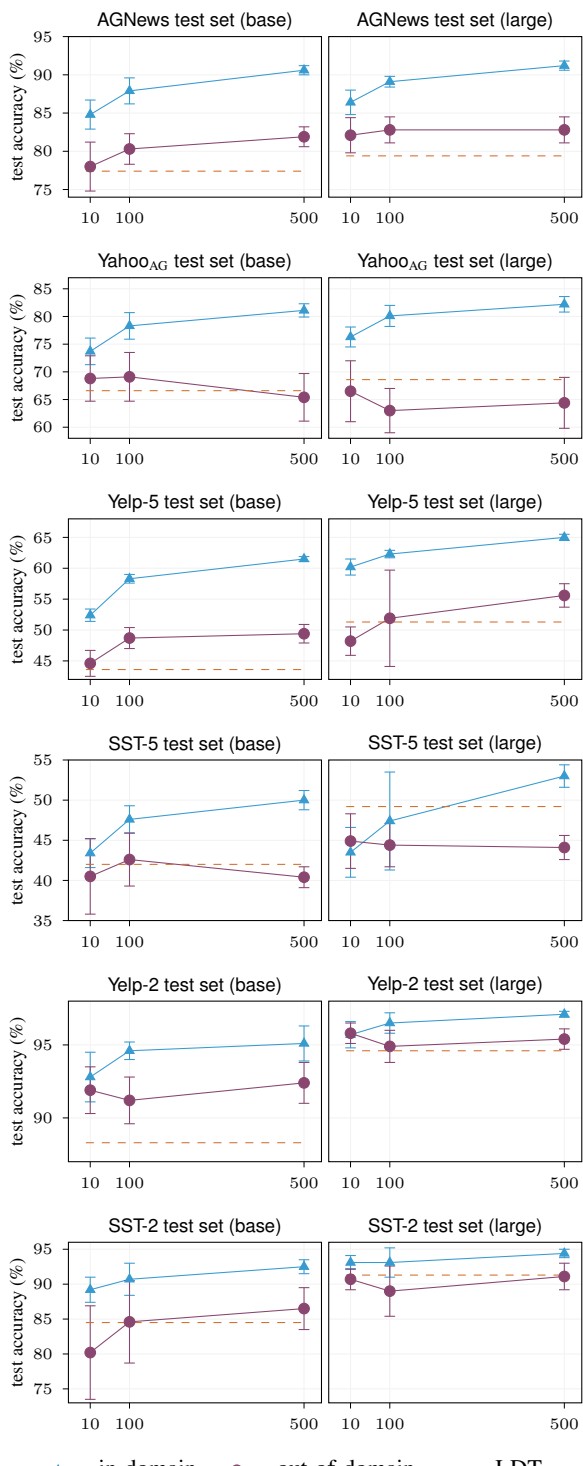

Figure 3: Domain transfer results, where X-axis depicts the number of training examples per label. "base/large" in parenthesis denotes RoBERTa-base/large.

| Label | Type | Training Data |
|---|---|---|
| World | terms | world \| country \| international \| politics |
| | Wiki. | In its most general sense, the term "world" refers to the totality of entities, to the whole of reality or to everything that is. |
| | dict. | humankind; the human race; humanity |
| Sports | terms | sport \| sports \| racing \| baseball |
| | Wiki. | Sport pertains to any form of competitive physical activity or game that aims to use, maintain or improve physical ability and skills while providing enjoyment to participants and, in some cases, entertainment to spectators. |
| | dict. | an athletic activity requiring skill or physical prowess and often of a competitive nature, as racing, baseball, tennis, golf, bowling, wrestling, boxing, hunting, fishing, etc. |
| Business | terms | business \| finance \| money \| trade |
| | Wiki. | Business is the activity of making one's living or making money by producing or buying and selling products (such as goods and services). |
| | dict. | the purchase and sale of goods in an attempt to make a profit. |
| Sci/Tech | terms | technology \| science \| computer \| biology |
| | Wiki. | Technology is the continually developing result of accumulated knowledge and application in all techniques, skills, methods, and processes used in industrial production and scientific research. |
| | dict. | the branch of knowledge that deals with the creation and use of technical means and their interrelation with life, society, and the environment, drawing upon such subjects as industrial arts, engineering, applied science, and pure science. |

Table 17: LABELDESC data for AGNews (and Yahoo_AG).

| Label | Type | Training Data |
|---|---|---|
| Very Negative | terms | awful \| terrible \| horrendous \| horrible \| dreadful |
| | sent. | It was $t$. \| A(n) $t$ experience. \| Just $t$. \| Overall, it was $t$. |
| Negative | terms | bad \| unpleasant \| unsatisfying \| lousy \| subpar |
| | sent. | It was $t$. \| A(n) $t$ experience. \| Just $t$. \| Overall, it was $t$. |
| Neutral | terms | okay \| mediocre \| decent \| average \| alright |
| | sent. | It was $t$. \| A(n) $t$ experience. \| Just $t$. \| Overall, it was $t$. |
| Positive | terms | good \| nice \| fine \| pleasant \| neat |
| | sent. | It was $t$. \| A(n) $t$ experience. \| Just $t$. \| Overall, it was $t$. |
| Very Positive | terms | great \| amazing \| excellent \| fantastic \| outstanding |
| | sent. | It was $t$. \| A(n) $t$ experience. \| Just $t$. \| Overall, it was $t$. |

Table 18: LABELDESC data for Yelp-5 and SST-5. "Sent." and "$t$" refer to hand-crafted sentence templates and terms, respectively.

| Label | Type | Training Data |
|---|---|---|
| Negative | terms | awful \| terrible \| horrendous \| horrible \| dreadful \| bad \| unpleasant \| unsatisfying \| lousy \| subpar |
| | sent. | It was $t$. \| A(n) $t$ experience. \| Just $t$. \| Overall, it was $t$. |
| Positive | terms | good \| nice \| fine \| pleasant \| neat \| great \| amazing \| excellent \| fantastic \| outstanding |
| | sent. | It was $t$. \| A(n) $t$ experience. \| Just $t$. \| Overall, it was $t$. |

Table 19: LABELDESC data for Yelp-2, SST-2, Amz-2 and IMDB.

| Label | Type | Training Data |
|---|---|---|
| Society & Culture | terms | society \| culture |
| | Wiki. | A society is a group of individuals involved in persistent social interaction, or a large social group sharing the same spatial or social territory, typically subject to the same political authority and dominant cultural expectations. \| Culture is an umbrella term which encompasses the social behavior, institutions, and norms found in human societies, as well as the knowledge, beliefs, arts, laws, customs, capabilities, and habits of the individuals in these groups. |
| | dict. | an organized group of persons associated together for religious, benevolent, cultural, scientific, political, patriotic, or other purposes. \| the behaviors and beliefs characteristic of a particular group of people, as a social, ethnic, professional, or age group (usually used in combination) |
| Science & Mathematics | terms | science \| mathematics |
| | Wiki. | Science is a systematic endeavor that builds and organizes knowledge in the form of testable explanations and predictions about the universe. \| Mathematics is an area of knowledge that includes such topics as numbers, formulas and related structures, shapes and the spaces in which they are contained, and quantities and their changes. |
| | dict. | a branch of knowledge or study dealing with a body of facts or truths systematically arranged and showing the operation of general laws \| the systematic treatment of magnitude, relationships between figures and forms, and relations between quantities expressed symbolically. |
| Health | terms | health \| fitness \| medical \| diet |
| | Wiki. | Health, according to the World Health Organization, is "a state of complete physical, mental and social well-being and not merely the absence of disease and infirmity". |
| | dict. | the general condition of the body or mind with reference to soundness and vigor |
| Education & Reference | terms | education \| reference |
| | Wiki. | Education is a purposeful activity directed at achieving certain aims, such as transmitting knowledge or fostering skills and character traits. \| Reference is a relationship between objects in which one object designates, or acts as a means by which to connect to or link to, another object. |
| | dict. | the act or process of imparting or acquiring general knowledge, developing the powers of reasoning and judgment, and generally of preparing oneself or others intellectually for mature life. \| a book or other source of useful facts or information, such as an encyclopedia, dictionary, etc. |
| Computers & Internet | terms | computer \| internet |
| | Wiki. | A computer is a digital electronic machine that can be programmed to carry out sequences of arithmetic or logical operations (computation) automatically. \| The Internet (or internet) is the global system of interconnected computer networks that uses the Internet protocol suite (TCP/IP) to communicate between networks and devices. |
| | dict. | a programmable electronic device designed to accept data, perform prescribed mathematical and logical operations at high speed, and display the results of these operations. Mainframes, desktop and laptop computers, tablets, and smartphones are some of the different types of computers. \| Usually the internet (except when used before a noun). a vast computer network linking smaller computer networks worldwide. The internet includes commercial, educational, governmental, and other networks, all of which use the same set of communications protocols |
| Sports | terms | sport \| sports \| racing \| baseball |
| | Wiki. | Sport pertains to any form of competitive physical activity or game that aims to use, maintain or improve physical ability and skills while providing enjoyment to participants and, in some cases, entertainment to spectators. |
| | dict. | an athletic activity requiring skill or physical prowess and often of a competitive nature, as racing, baseball, tennis, golf, bowling, wrestling, boxing, hunting, fishing, etc. |
| Business & Finance | terms | business \| finance |
| | Wiki. | Business is the activity of making one's living or making money by producing or buying and selling products (such as goods and services). \| Finance is the study and discipline of money, currency and capital assets. |
| | dict. | the purchase and sale of goods in an attempt to make a profit. \| the management of revenues; the conduct or transaction of money matters generally, especially those affecting the public, as in the fields of banking and investment. |
| Entertainment & Music | terms | entertainment \| music |
| | Wiki. | Entertainment is a form of activity that holds the attention and interest of an audience or gives pleasure and delight. \| Music is generally defined as the art of arranging sound to create some combination of form, harmony, melody, rhythm or otherwise expressive content. |
| | dict. | the act of entertaining; agreeable occupation for the mind; diversion; amusement \| an art of sound in time that expresses ideas and emotions in significant forms through the elements of rhythm, melody, harmony, and color. |
| Family & Relationships | terms | family \| relationship |
| | Wiki. | Family is a group of people related either by consanguinity (by recognized birth) or affinity (by marriage or other relationship). \| The concept of interpersonal relationship involves social associations, connections, or affiliations between two or more people. |
| | dict. | a basic social unit consisting of parents and their children, considered as a group, whether dwelling together or not; a social unit consisting of one or more adults together with the children they care for. \| an emotional or other connection between people |
| Politics & Government | terms | politics \| government |
| | Wiki. | Politics is the set of activities that are associated with making decisions in groups, or other forms of power relations among individuals, such as the distribution of resources or status. \| A government is the system or group of people governing an organized community, generally a state. |
| | dict. | the science or art of political government. \| the political direction and control exercised over the actions of the members, citizens, or inhabitants of communities, societies, and states; direction of the affairs of a state, community, etc.; political administration |

Table 20: LABELDESC data for Yahoo Answers.

| Label | Type | Training Data |
|---|---|---|
| Company | terms | company \| firm \| corporation \| business |
| | Wiki. | A company, abbreviated as co., is a legal entity representing an association of people, whether natural, legal or a mixture of both, with a specific objective. |
| | dict. | a number of persons united or incorporated for joint action, especially for business |
| Educational Institution | terms | educational institution \| university \| college \| school |
| | Wiki. | An educational institution is a place where people of different ages gain an education, including preschools, childcare, primary-elementary schools, secondary-high schools, and universities. |
| | dict. | an institution for instruction in a particular skill or field. |
| Artist | terms | artist \| writer \| actor \| singer |
| | Wiki. | An artist is a person engaged in an activity related to creating art, practicing the arts, or demonstrating an art. |
| | dict. | a person who produces works in any of the arts that are primarily subject to aesthetic criteria. |
| Athlete | terms | athlete \| sports \| footballer \| weightlifter |
| | Wiki. | An athlete (also sportsman or sportswoman) is a person who competes in one or more sports that involve physical strength, speed, or endurance. |
| | dict. | a person trained or gifted in exercises or contests involving physical agility, stamina, or strength; a participant in a sport, exercise, or game requiring physical skill. |
| Office Holder | terms | office-holder \| politics \| mayor \| president |
| | Wiki. | A person who's been appointed to a position by a company or organisation but doesn't have a contract or receive regular payment may be an office-holder. |
| | dict. | a person filling a governmental position; public official. |
| Mean of Transportation | terms | mean of transportation \| car \| bus \| train |
| | Wiki. | Transport (in British English), or transportation (in American English), is the intentional movement of humans, animals, and goods from one location to another. |
| | dict. | a means of transporting or conveying, as a truck or bus. |
| Building | terms | building \| apartment \| skyscraper \| tower |
| | Wiki. | A building or edifice, is an enclosed structure with a roof and walls standing more or less permanently in one place, such as a house or factory (although there's also portable buildings). |
| | dict. | a relatively permanent enclosed construction over a plot of land, having a roof and usually windows and often more than one level, used for any of a wide variety of activities, as living, entertaining, or manufacturing. |
| Natural Place | terms | natural place \| forest \| mountain \| river |
| | Wiki. | The natural environment or natural world encompasses all living and non-living things occurring naturally, meaning in this case not artificial. |
| | dict. | existing in or formed by nature (opposed to artificial) |
| Village | terms | village \| town \| countryside \| rural |
| | Wiki. | A village is a clustered human settlement or community, larger than a hamlet but smaller than a town (although the word is often used to describe both hamlets and smaller towns), with a population typically ranging from a few hundred to a few thousand. |
| | dict. | a small community or group of houses in a rural area, larger than a hamlet and usually smaller than a town, and sometimes (as in parts of the U.S.) incorporated as a municipality. |
| Animal | terms | animal \| insect \| bird \| fish |
| | Wiki. | Animals are multicellular, eukaryotic organisms in the biological kingdom Animalia. |
| | dict. | any member of the kingdom Animalia, comprising multicellular organisms that have a well-defined shape and usually limited growth, can move voluntarily, actively acquire food and digest it internally, and have sensory and nervous systems that allow them to respond rapidly to stimuli: some classification schemes also include protozoa and certain other single-celled eukaryotes that have motility and animallike nutritional modes. |
| Plant | terms | plant \| flower \| tree \| grass |
| | Wiki. | Plants are predominantly photosynthetic eukaryotes, forming the kingdom Plantae. |
| | dict. | Botany. any member of the kingdom Plantae, comprising multicellular organisms that typically produce their own food from inorganic matter by the process of photosynthesis and that have more or less rigid cell walls containing cellulose, including vascular plants, mosses, liverworts, and hornworts: some classification schemes may include fungi, algae, bacteria, and certain single-celled eukaryotes that have plantlike qualities, as rigid cell walls or the use of photosynthesis. |
| Album | terms | album \| soundtrack \| mixtape \| CD |
| | Wiki. | An album is a collection of audio recordings issued on compact disc (CD), vinyl, audio tape, or another medium such as digital distribution. |
| | dict. | a record or set of records containing several musical selections, a complete play or opera, etc. |
| Film | terms | film \| movie \| comedy \| drama |
| | Wiki. | A film – also called a movie, motion picture, moving picture, picture, photoplay or (slang) flick – is a work of visual art that simulates experiences and otherwise communicates ideas, stories, perceptions, feelings, beauty, or atmosphere through the use of moving images. |
| | dict. | a sequence of consecutive still images recorded in a series to be viewed on a screen in such rapid succession as to give the illusion of natural movement; motion picture. |
| Written Work | terms | written work \| novel \| newspaper \| book |
| | Wiki. | A book is a medium for recording information in the form of writing or images, typically composed of many pages (made of papyrus, parchment, vellum, or paper) bound together and protected by a cover. |
| | dict. | a handwritten or printed work of fiction or nonfiction, usually on sheets of paper fastened or bound together within covers. |

Table 21: LABELDESC data for DBPedia.

| dataset | RoBERTa | label | precision(%) | | recall(%) | | F1(%) | |
|---|---|---|---|---|---|---|---|---|
| | | | zero-shot | LDT | zero-shot | LDT | zero-shot | LDT |
| AGNews | base | World | $58.7_{\pm12.8}$ | $80.2_{\pm7.3}$ | $29.1_{\pm27.8}$ | $62.0_{\pm18.2}$ | $33.7_{\pm21.2}$ | $68.0_{\pm12.0}$ |
| | | Business | $60.6_{\pm8.1}$ | $71.0_{\pm6.3}$ | $66.9_{\pm14.0}$ | $77.0_{\pm4.6}$ | $63.0_{\pm9.7}$ | $73.7_{\pm3.9}$ |
| | | Sports | $72.9_{\pm14.7}$ | $94.1_{\pm1.5}$ | $92.5_{\pm9.6}$ | $94.4_{\pm6.2}$ | $80.1_{\pm8.7}$ | $94.1_{\pm3.3}$ |
| | | Sci/Tech | $65.3_{\pm15.8}$ | $69.9_{\pm9.2}$ | $62.4_{\pm14.8}$ | $76.4_{\pm6.0}$ | $60.6_{\pm8.1}$ | $72.4_{\pm4.3}$ |
| | large | World | $81.6_{\pm10.0}$ | $78.8_{\pm6.3}$ | $53.1_{\pm21.3}$ | $84.1_{\pm6.8}$ | $61.5_{\pm15.1}$ | $81.0_{\pm4.3}$ |
| | | Business | $53.1_{\pm13.7}$ | $67.4_{\pm9.9}$ | $84.6_{\pm9.2}$ | $86.4_{\pm5.8}$ | $63.6_{\pm7.1}$ | $74.9_{\pm4.7}$ |
| | | Sports | $86.8_{\pm5.3}$ | $95.6_{\pm1.2}$ | $90.4_{\pm8.2}$ | $90.2_{\pm7.8}$ | $88.2_{\pm3.9}$ | $92.7_{\pm4.5}$ |
| | | Sci/Tech | $75.0_{\pm12.3}$ | $87.0_{\pm4.8}$ | $44.1_{\pm11.9}$ | $57.0_{\pm13.2}$ | $55.0_{\pm11.4}$ | $67.8_{\pm9.3}$ |
| Yahoo$_{AG}$ | base | World | $44.8_{\pm6.9}$ | $64.4_{\pm4.4}$ | $34.9_{\pm25.3}$ | $62.9_{\pm12.6}$ | $35.8_{\pm13.8}$ | $62.7_{\pm5.3}$ |
| | | Business | $45.2_{\pm4.3}$ | $53.1_{\pm7.3}$ | $48.7_{\pm6.8}$ | $52.9_{\pm6.2}$ | $46.5_{\pm3.2}$ | $52.3_{\pm2.7}$ |
| | | Sports | $49.4_{\pm21.5}$ | $86.7_{\pm5.8}$ | $83.7_{\pm17.3}$ | $78.7_{\pm7.6}$ | $57.1_{\pm8.4}$ | $82.1_{\pm3.9}$ |
| | | Sci/Tech | $72.9_{\pm14.1}$ | $70.0_{\pm8.5}$ | $45.3_{\pm15.1}$ | $71.2_{\pm7.1}$ | $52.7_{\pm12.7}$ | $69.9_{\pm3.0}$ |
| | large | World | $63.4_{\pm9.1}$ | $77.2_{\pm5.6}$ | $25.6_{\pm20.8}$ | $53.8_{\pm12.0}$ | $32.6_{\pm15.6}$ | $62.5_{\pm7.8}$ |
| | | Business | $33.4_{\pm8.0}$ | $47.3_{\pm8.6}$ | $70.2_{\pm8.9}$ | $63.7_{\pm8.3}$ | $44.1_{\pm4.1}$ | $53.1_{\pm3.0}$ |
| | | Sports | $61.2_{\pm15.7}$ | $88.3_{\pm4.3}$ | $85.9_{\pm10.0}$ | $82.9_{\pm5.7}$ | $69.4_{\pm6.5}$ | $85.3_{\pm2.5}$ |
| | | Sci/Tech | $72.9_{\pm8.6}$ | $72.4_{\pm9.1}$ | $50.9_{\pm8.9}$ | $78.7_{\pm9.3}$ | $59.5_{\pm7.5}$ | $74.5_{\pm3.8}$ |
| Yelp-5 | base | Very Negative | $67.1_{\pm32.8}$ | $68.6_{\pm4.9}$ | $5.3_{\pm9.6}$ | $37.8_{\pm14.5}$ | $8.3_{\pm13.3}$ | $46.4_{\pm12.4}$ |
| | | Negative | $34.5_{\pm4.4}$ | $41.5_{\pm2.3}$ | $75.2_{\pm16.7}$ | $55.7_{\pm10.9}$ | $46.2_{\pm3.7}$ | $46.9_{\pm2.8}$ |
| | | Neutral | $34.2_{\pm20.7}$ | $53.0_{\pm2.9}$ | $2.5_{\pm3.3}$ | $14.1_{\pm5.1}$ | $4.3_{\pm5.5}$ | $21.7_{\pm6.1}$ |
| | | Positive | $34.4_{\pm6.1}$ | $29.2_{\pm2.3}$ | $50.8_{\pm23.2}$ | $16.3_{\pm5.0}$ | $38.4_{\pm10.1}$ | $20.5_{\pm4.3}$ |
| | | Very Positive | $59.6_{\pm11.8}$ | $42.0_{\pm2.1}$ | $56.2_{\pm25.0}$ | $94.0_{\pm1.8}$ | $52.4_{\pm11.8}$ | $58.0_{\pm1.8}$ |
| | large | Very Negative | $78.6_{\pm20.2}$ | $82.9_{\pm2.9}$ | $12.8_{\pm21.4}$ | $34.3_{\pm11.3}$ | $16.2_{\pm20.5}$ | $47.2_{\pm11.9}$ |
| | | Negative | $39.4_{\pm5.2}$ | $44.7_{\pm2.6}$ | $55.0_{\pm23.9}$ | $75.8_{\pm5.1}$ | $43.2_{\pm12.3}$ | $56.0_{\pm1.1}$ |
| | | Neutral | $39.2_{\pm22.8}$ | $64.2_{\pm2.8}$ | $4.4_{\pm7.3}$ | $20.7_{\pm8.5}$ | $7.0_{\pm10.5}$ | $30.2_{\pm9.9}$ |
| | | Positive | $31.5_{\pm5.6}$ | $38.5_{\pm3.9}$ | $84.8_{\pm9.9}$ | $36.8_{\pm13.4}$ | $45.4_{\pm5.6}$ | $36.8_{\pm7.7}$ |
| | | Very Positive | $72.1_{\pm8.9}$ | $56.2_{\pm5.3}$ | $36.7_{\pm22.8}$ | $88.8_{\pm8.6}$ | $44.2_{\pm22.2}$ | $68.2_{\pm2.4}$ |
| SST-5 | base | Very Negative | $13.7_{\pm15.1}$ | $26.9_{\pm3.1}$ | $4.4_{\pm8.1}$ | $24.2_{\pm7.4}$ | $5.7_{\pm8.6}$ | $24.7_{\pm4.3}$ |
| | | Negative | $45.0_{\pm7.7}$ | $53.2_{\pm1.6}$ | $60.3_{\pm29.8}$ | $58.0_{\pm7.4}$ | $46.4_{\pm8.6}$ | $55.2_{\pm3.0}$ |
| | | Neutral | $8.9_{\pm11.5}$ | $26.0_{\pm3.3}$ | $0.7_{\pm1.8}$ | $11.7_{\pm7.6}$ | $1.2_{\pm2.9}$ | $14.9_{\pm6.6}$ |
| | | Positive | $33.0_{\pm5.4}$ | $39.5_{\pm2.4}$ | $57.5_{\pm27.5}$ | $30.1_{\pm9.4}$ | $38.1_{\pm11.4}$ | $33.3_{\pm6.8}$ |
| | | Very Positive | $45.9_{\pm21.1}$ | $42.5_{\pm3.3}$ | $24.3_{\pm27.2}$ | $73.6_{\pm7.0}$ | $22.9_{\pm17.2}$ | $53.6_{\pm1.5}$ |
| | large | Very Negative | $21.1_{\pm20.5}$ | $35.9_{\pm6.0}$ | $13.6_{\pm23.6}$ | $21.2_{\pm7.4}$ | $11.2_{\pm14.9}$ | $25.8_{\pm5.7}$ |
| | | Negative | $51.7_{\pm4.7}$ | $54.5_{\pm1.4}$ | $36.7_{\pm26.8}$ | $73.7_{\pm6.5}$ | $37.6_{\pm21.2}$ | $62.5_{\pm2.0}$ |
| | | Neutral | $7.5_{\pm13.0}$ | $32.3_{\pm6.8}$ | $0.7_{\pm1.7}$ | $6.7_{\pm3.8}$ | $1.2_{\pm2.9}$ | $10.8_{\pm5.5}$ |
| | | Positive | $31.3_{\pm6.1}$ | $46.2_{\pm2.6}$ | $91.5_{\pm9.5}$ | $52.6_{\pm12.7}$ | $46.0_{\pm5.8}$ | $48.2_{\pm4.9}$ |
| | | Very Positive | $66.8_{\pm25.6}$ | $53.4_{\pm6.4}$ | $8.3_{\pm12.0}$ | $67.2_{\pm13.5}$ | $12.1_{\pm15.0}$ | $58.0_{\pm4.0}$ |
| Yelp-2 | base | Negative | $91.8_{\pm26.4}$ | $97.0_{\pm0.9}$ | $27.6_{\pm21.7}$ | $79.1_{\pm5.7}$ | $38.7_{\pm27.9}$ | $87.0_{\pm3.2}$ |
| | | Positive | $58.8_{\pm7.3}$ | $82.5_{\pm3.8}$ | $99.7_{\pm0.3}$ | $97.5_{\pm0.9}$ | $73.7_{\pm5.7}$ | $89.3_{\pm1.9}$ |
| | large | Negative | $99.5_{\pm0.4}$ | $97.6_{\pm0.9}$ | $41.4_{\pm31.7}$ | $91.6_{\pm4.1}$ | $51.5_{\pm33.7}$ | $94.4_{\pm2.1}$ |
| | | Positive | $65.5_{\pm13.5}$ | $92.2_{\pm3.3}$ | $99.7_{\pm0.3}$ | $97.7_{\pm0.9}$ | $78.3_{\pm9.6}$ | $94.8_{\pm1.5}$ |
| SST-2 | base | Negative | $85.5_{\pm25.1}$ | $81.8_{\pm4.0}$ | $28.9_{\pm25.3}$ | $89.1_{\pm3.9}$ | $37.7_{\pm29.6}$ | $85.2_{\pm1.9}$ |
| | | Positive | $58.8_{\pm8.4}$ | $88.3_{\pm3.1}$ | $96.5_{\pm3.6}$ | $79.9_{\pm5.9}$ | $72.6_{\pm5.4}$ | $83.7_{\pm2.9}$ |
| | large | Negative | $83.2_{\pm35.3}$ | $94.5_{\pm2.3}$ | $28.8_{\pm29.9}$ | $88.0_{\pm5.4}$ | $36.5_{\pm35.1}$ | $91.0_{\pm2.6}$ |
| | | Positive | $59.9_{\pm11.2}$ | $89.0_{\pm3.9}$ | $98.8_{\pm1.6}$ | $94.7_{\pm2.5}$ | $74.0_{\pm7.9}$ | $91.7_{\pm1.7}$ |

Table 22: Precision, recall, and F1 score for zero-shot and LABELDESCTRAINING (for brevity, LDT in header).

| dataset | RoBERTa | label | precision(%) zero-shot | precision(%) LDT | recall(%) zero-shot | recall(%) LDT | F1(%) zero-shot | F1(%) LDT |
|---|---|---|---|---|---|---|---|---|
| Yahoo | base | Society & Culture | 27.2±9.1 | 39.0±6.1 | 2.7±2.7 | 8.8±3.2 | 4.5±4.0 | 14.1±4.2 |
| | | Science & Mathematics | 40.0±14.0 | 60.4±5.2 | 68.5±13.3 | 58.8±5.0 | 47.4±11.2 | 59.3±2.4 |
| | | Health | 52.1±6.9 | 59.1±6.9 | 74.9±12.0 | 77.6±4.2 | 60.4±4.7 | 66.6±3.4 |
| | | Education & Reference | 39.8±12.2 | 46.5±4.9 | 22.0±12.3 | 36.8±4.9 | 24.6±10.4 | 40.6±2.6 |
| | | Computers & Internet | 80.1±9.2 | 69.1±3.9 | 19.2±11.5 | 78.2±5.0 | 29.6±14.3 | 73.1±1.9 |
| | | Sports | 70.7±21.6 | 81.9±6.8 | 65.9±21.8 | 81.7±2.3 | 61.8±11.5 | 81.5±3.0 |
| | | Business & Finance | 41.4±9.9 | 54.4±3.0 | 34.9±8.9 | 44.7±3.2 | 36.1±4.2 | 48.9±1.7 |
| | | Entertainment & Music | 49.4±12.3 | 62.5±5.6 | 39.5±20.3 | 58.0±4.4 | 38.7±13.5 | 59.8±2.1 |
| | | Family & Relationships | 70.0±9.5 | 47.2±3.9 | 30.2±14.6 | 81.2±9.8 | 40.2±14.2 | 59.2±2.5 |
| | | Politics & Government | 46.5±20.5 | 64.5±7.1 | 57.4±27.1 | 62.2±7.4 | 40.6±17.8 | 62.6±2.4 |
| | large | Society & Culture | 35.3±9.4 | 46.1±7.0 | 2.2±4.0 | 9.6±4.8 | 3.7±5.7 | 15.5±6.4 |
| | | Science & Mathematics | 48.6±11.8 | 65.7±5.9 | 67.1±11.1 | 66.2±4.5 | 54.3±6.6 | 65.6±2.0 |
| | | Health | 54.0±10.7 | 65.5±3.7 | 82.5±11.1 | 81.2±3.4 | 63.7±3.7 | 72.3±1.6 |
| | | Education & Reference | 33.2±14.9 | 38.3±7.5 | 34.1±11.1 | 46.9±6.9 | 30.2±9.5 | 41.4±4.4 |
| | | Computers & Internet | 81.8±9.5 | 80.2±3.8 | 36.2±16.9 | 75.1±5.3 | 47.3±12.6 | 77.3±2.1 |
| | | Sports | 75.8±15.9 | 87.8±5.5 | 74.9±17.5 | 83.3±3.9 | 72.1±11.0 | 85.3±2.3 |
| | | Business & Finance | 37.2±9.4 | 56.2±5.2 | 46.5±14.4 | 43.9±5.7 | 38.7±6.0 | 48.8±2.4 |
| | | Entertainment & Music | 52.7±8.2 | 64.5±4.9 | 38.3±22.0 | 58.9±6.7 | 39.7±17.1 | 61.2±3.6 |
| | | Family & Relationships | 71.2±8.4 | 47.5±5.0 | 47.5±26.5 | 85.0±8.8 | 50.8±25.8 | 60.4±3.9 |
| | | Politics & Government | 57.5±15.9 | 71.8±5.0 | 48.0±20.5 | 57.5±4.5 | 46.1±16.1 | 63.5±1.4 |

Table 23: Precision, recall, and F1 score for zero-shot and LABELDESCTRAINING.

| AGNews | | zero-shot | In-domain | | | Out-of-domain | | | LDT |
|---|---|---|---|---|---|---|---|---|---|
| | | | 10 | 100 | 500 | 10 | 100 | 500 | |
| Prec.-b | World | 58.7±12.8 | 84.7±2.6 | 88.5±4.2 | 93.0±2.6 | 77.4±8.6 | 81.2±5.5 | 79.8±6.8 | 80.2±7.3 |
| | Business | 60.6±8.1 | 80.9±4.6 | 83.0±3.8 | 86.9±3.0 | 75.2±6.1 | 72.8±7.5 | 73.1±4.4 | 71.0±6.3 |
| | Sports | 72.9±14.7 | 94.7±1.3 | 95.7±1.4 | 96.6±0.6 | 91.6±3.2 | 92.1±3.0 | 94.4±0.7 | 94.1±1.5 |
| | Sci/Tech | 65.3±15.8 | 79.6±6.6 | 85.8±4.5 | 86.6±2.6 | 72.6±9.7 | 78.6±7.0 | 83.2±4.9 | 69.9±9.2 |
| Rec.-b | World | 29.1±27.8 | 85.7±3.6 | 87.1±3.0 | 88.7±3.4 | 78.7±10.4 | 78.8±5.1 | 81.0±5.2 | 62.0±18.2 |
| | Business | 66.9±14.0 | 77.4±7.4 | 83.6±7.5 | 86.4±2.7 | 63.3±14.5 | 76.2±11.4 | 81.7±5.6 | 77.0±4.6 |
| | Sports | 92.5±9.6 | 98.1±0.8 | 96.9±2.6 | 98.2±0.5 | 97.4±2.2 | 98.4±0.6 | 97.9±0.8 | 94.4±6.2 |
| | Sci/Tech | 62.4±14.8 | 78.0±5.8 | 84.2±4.8 | 89.2±3.3 | 72.5±9.9 | 67.9±10.1 | 66.8±8.1 | 76.4±6.0 |
| F1-b | World | 33.7±21.2 | 85.1±1.2 | 87.6±1.9 | 90.7±1.1 | 77.0±4.4 | 79.7±2.2 | 80.0±1.9 | 68.0±12.0 |
| | Business | 63.0±9.7 | 78.8±3.7 | 82.9±3.4 | 86.6±0.8 | 67.3±8.5 | 73.3±4.8 | 76.9±1.6 | 73.7±3.9 |
| | Sports | 80.1±8.7 | 96.4±0.4 | 96.2±1.3 | 97.4±0.3 | 94.4±1.7 | 95.1±1.6 | 96.1±0.3 | 94.1±3.3 |
| | Sci/Tech | 60.6±8.1 | 78.4±3.3 | 84.8±2.1 | 87.8±0.9 | 71.4±3.7 | 71.9±4.5 | 73.6±3.6 | 72.4±4.3 |
| Prec.-l | World | 81.6±10.0 | 84.7±2.3 | 89.0±2.0 | 93.6±2.8 | 77.4±6.7 | 83.9±4.2 | 81.4±6.2 | 78.8±6.3 |
| | Business | 53.1±13.7 | 83.8±3.2 | 83.7±3.0 | 88.0±2.4 | 79.5±5.8 | 74.0±5.7 | 72.4±5.3 | 67.4±9.9 |
| | Sports | 86.8±5.3 | 95.2±1.1 | 96.3±0.6 | 96.9±0.9 | 93.1±3.3 | 93.0±2.3 | 94.6±0.9 | 95.6±1.2 |
| | Sci/Tech | 75.0±12.3 | 82.4±5.7 | 87.8±2.4 | 86.8±2.5 | 80.6±4.8 | 82.6±5.7 | 87.1±3.8 | 87.0±4.8 |
| Rec.-l | World | 53.1±21.3 | 89.0±2.1 | 89.0±1.3 | 89.2±3.6 | 86.7±4.7 | 82.9±3.4 | 82.4±5.0 | 84.1±6.8 |
| | Business | 84.6±9.2 | 78.6±6.8 | 85.8±3.6 | 86.8±2.2 | 70.4±9.3 | 79.3±7.3 | 83.6±4.3 | 86.4±5.8 |
| | Sports | 90.4±8.2 | 97.8±0.9 | 97.9±1.5 | 98.5±0.8 | 98.5±0.8 | 98.5±0.7 | 97.7±1.8 | 90.2±7.8 |
| | Sci/Tech | 44.1±11.9 | 80.2±4.1 | 83.6±4.5 | 90.2±3.0 | 73.0±9.1 | 70.4±9.9 | 67.5±9.3 | 57.0±13.2 |
| F1-l | World | 61.5±15.1 | 86.7±0.8 | 89.0±0.8 | 91.3±1.2 | 81.4±3.1 | 83.2±1.4 | 81.5±1.7 | 81.0±4.3 |
| | Business | 63.6±7.1 | 80.9±3.9 | 84.6±1.0 | 87.3±0.7 | 74.0±4.7 | 76.1±2.4 | 77.3±2.0 | 74.9±4.7 |
| | Sports | 88.2±3.9 | 96.5±0.5 | 97.1±0.5 | 97.7±0.3 | 95.7±1.8 | 95.7±1.1 | 96.1±0.8 | 92.7±4.5 |
| | Sci/Tech | 55.0±11.4 | 81.0±2.3 | 85.5±1.5 | 88.4±0.7 | 76.0±4.0 | 75.2±5.1 | 75.5±5.4 | 67.8±9.3 |

Table 24: Precision, recall, and F1 for AGNews, where 'b' refers to RoBERTa-base, 'l' refers to RoBERTa-large.

| Yahoo$_{AG}$ | | zero-shot | In-domain | | | Out-of-domain | | | LDT |
|---|---|---|---|---|---|---|---|---|---|
| | | | 10 | 100 | 500 | 10 | 100 | 500 | |
| Prec.-b | World | 44.8$_{\pm6.9}$ | 80.4$_{\pm5.1}$ | 82.8$_{\pm2.9}$ | 85.4$_{\pm3.0}$ | 76.0$_{\pm6.6}$ | 84.3$_{\pm5.0}$ | 85.5$_{\pm4.3}$ | 64.4$_{\pm4.4}$ |
| | Business | 45.2$_{\pm4.3}$ | 48.4$_{\pm6.0}$ | 53.3$_{\pm7.9}$ | 56.2$_{\pm5.2}$ | 63.5$_{\pm11.2}$ | 58.1$_{\pm8.6}$ | 50.2$_{\pm7.5}$ | 53.1$_{\pm7.3}$ |
| | Sports | 49.4$_{\pm21.5}$ | 83.8$_{\pm6.7}$ | 86.6$_{\pm6.1}$ | 91.2$_{\pm2.5}$ | 81.9$_{\pm13.8}$ | 89.6$_{\pm5.3}$ | 88.9$_{\pm5.9}$ | 86.7$_{\pm5.8}$ |
| | Sci/Tech | 72.9$_{\pm14.1}$ | 82.4$_{\pm5.0}$ | 88.0$_{\pm3.4}$ | 88.9$_{\pm2.0}$ | 65.1$_{\pm7.7}$ | 62.3$_{\pm7.8}$ | 60.0$_{\pm6.7}$ | 70.0$_{\pm8.5}$ |
| Rec.-b | World | 34.9$_{\pm25.3}$ | 67.4$_{\pm10.2}$ | 75.9$_{\pm5.8}$ | 77.7$_{\pm5.4}$ | 62.5$_{\pm16.2}$ | 50.1$_{\pm14.2}$ | 37.6$_{\pm13.5}$ | 62.9$_{\pm12.6}$ |
| | Business | 48.7$_{\pm6.8}$ | 61.7$_{\pm9.7}$ | 63.6$_{\pm9.8}$ | 68.9$_{\pm5.7}$ | 34.2$_{\pm16.7}$ | 46.1$_{\pm10.3}$ | 48.3$_{\pm7.6}$ | 52.9$_{\pm6.2}$ |
| | Sports | 83.7$_{\pm17.3}$ | 85.7$_{\pm5.7}$ | 91.0$_{\pm2.2}$ | 91.1$_{\pm1.7}$ | 85.7$_{\pm5.6}$ | 82.4$_{\pm6.4}$ | 80.8$_{\pm3.6}$ | 78.7$_{\pm7.6}$ |
| | Sci/Tech | 45.3$_{\pm15.1}$ | 80.2$_{\pm5.6}$ | 81.7$_{\pm6.3}$ | 85.7$_{\pm2.9}$ | 84.0$_{\pm5.1}$ | 92.8$_{\pm3.4}$ | 94.2$_{\pm2.7}$ | 71.2$_{\pm7.1}$ |
| F1-b | World | 35.8$_{\pm13.8}$ | 72.5$_{\pm4.9}$ | 79.0$_{\pm2.5}$ | 81.2$_{\pm1.7}$ | 66.5$_{\pm9.4}$ | 61.4$_{\pm9.4}$ | 50.5$_{\pm12.2}$ | 62.7$_{\pm5.3}$ |
| | Business | 46.5$_{\pm3.2}$ | 53.3$_{\pm3.2}$ | 56.7$_{\pm3.3}$ | 61.4$_{\pm1.1}$ | 40.6$_{\pm15.0}$ | 49.8$_{\pm7.3}$ | 48.2$_{\pm2.9}$ | 52.3$_{\pm2.7}$ |
| | Sports | 57.1$_{\pm8.4}$ | 84.4$_{\pm2.5}$ | 88.6$_{\pm2.8}$ | 91.1$_{\pm0.7}$ | 82.6$_{\pm6.6}$ | 85.5$_{\pm3.4}$ | 84.4$_{\pm2.5}$ | 82.1$_{\pm3.9}$ |
| | Sci/Tech | 52.7$_{\pm12.7}$ | 81.0$_{\pm1.9}$ | 84.5$_{\pm2.6}$ | 87.2$_{\pm0.8}$ | 72.9$_{\pm3.7}$ | 74.1$_{\pm4.7}$ | 73.0$_{\pm4.3}$ | 69.9$_{\pm3.0}$ |
| Prec.-l | World | 63.4$_{\pm9.1}$ | 81.1$_{\pm4.0}$ | 84.2$_{\pm3.6}$ | 85.8$_{\pm3.1}$ | 83.6$_{\pm6.0}$ | 90.0$_{\pm2.6}$ | 89.3$_{\pm3.8}$ | 77.2$_{\pm5.6}$ |
| | Business | 33.4$_{\pm8.0}$ | 50.8$_{\pm4.8}$ | 55.1$_{\pm6.4}$ | 57.6$_{\pm5.3}$ | 68.3$_{\pm11.6}$ | 60.7$_{\pm6.4}$ | 54.8$_{\pm7.5}$ | 47.3$_{\pm8.6}$ |
| | Sports | 61.2$_{\pm15.7}$ | 87.0$_{\pm5.6}$ | 88.6$_{\pm5.5}$ | 93.1$_{\pm2.2}$ | 81.5$_{\pm11.8}$ | 86.3$_{\pm5.4}$ | 89.2$_{\pm7.6}$ | 88.3$_{\pm4.3}$ |
| | Sci/Tech | 72.9$_{\pm8.6}$ | 86.2$_{\pm2.7}$ | 89.5$_{\pm2.3}$ | 90.1$_{\pm2.4}$ | 58.6$_{\pm8.7}$ | 52.6$_{\pm3.9}$ | 56.7$_{\pm7.7}$ | 72.4$_{\pm9.1}$ |
| Rec.-l | World | 25.6$_{\pm20.8}$ | 69.4$_{\pm7.3}$ | 77.5$_{\pm7.0}$ | 79.7$_{\pm5.6}$ | 51.1$_{\pm18.2}$ | 31.1$_{\pm10.3}$ | 33.4$_{\pm13.0}$ | 53.8$_{\pm12.0}$ |
| | Business | 70.2$_{\pm8.9}$ | 68.3$_{\pm5.0}$ | 66.9$_{\pm7.4}$ | 70.1$_{\pm5.7}$ | 28.8$_{\pm16.3}$ | 38.4$_{\pm8.1}$ | 44.6$_{\pm9.6}$ | 63.7$_{\pm8.3}$ |
| | Sports | 85.9$_{\pm10.0}$ | 88.8$_{\pm4.6}$ | 92.4$_{\pm2.2}$ | 91.7$_{\pm2.4}$ | 87.9$_{\pm4.8}$ | 86.4$_{\pm2.6}$ | 83.8$_{\pm4.5}$ | 82.9$_{\pm5.7}$ |
| | Sci/Tech | 50.9$_{\pm8.9}$ | 81.0$_{\pm3.9}$ | 83.0$_{\pm4.0}$ | 86.1$_{\pm4.6}$ | 90.2$_{\pm4.9}$ | 95.6$_{\pm1.2}$ | 95.6$_{\pm3.0}$ | 78.7$_{\pm9.3}$ |
| F1-l | World | 32.6$_{\pm15.6}$ | 74.5$_{\pm3.6}$ | 80.4$_{\pm2.8}$ | 82.4$_{\pm1.9}$ | 60.8$_{\pm12.7}$ | 45.2$_{\pm11.3}$ | 46.9$_{\pm13.1}$ | 62.5$_{\pm7.8}$ |
| | Business | 44.1$_{\pm4.1}$ | 57.9$_{\pm1.8}$ | 59.7$_{\pm2.0}$ | 62.8$_{\pm1.3}$ | 36.6$_{\pm15.1}$ | 46.2$_{\pm6.1}$ | 47.9$_{\pm4.6}$ | 53.1$_{\pm3.0}$ |
| | Sports | 69.4$_{\pm6.5}$ | 87.6$_{\pm2.3}$ | 90.3$_{\pm2.4}$ | 92.3$_{\pm0.8}$ | 83.8$_{\pm5.2}$ | 86.2$_{\pm2.2}$ | 86.0$_{\pm3.3}$ | 85.3$_{\pm2.5}$ |
| | Sci/Tech | 59.5$_{\pm7.5}$ | 83.4$_{\pm1.4}$ | 86.0$_{\pm1.5}$ | 87.9$_{\pm1.7}$ | 70.4$_{\pm4.9}$ | 67.8$_{\pm3.0}$ | 70.8$_{\pm5.0}$ | 74.5$_{\pm3.8}$ |

Table 25: Precision, recall, and F1 for Yahoo$_{AG}$, where 'b' refers to RoBERTa-base, 'l' refers to RoBERTa-large.

| Yelp-5 | | zero-shot | In-domain | | | Out-of-domain | | | LDT |
|---|---|---|---|---|---|---|---|---|---|
| | | | 10 | 100 | 500 | 10 | 100 | 500 | |
| Prec.-b | Very Negative | $67.1_{\pm32.8}$ | $65.5_{\pm7.0}$ | $71.1_{\pm2.4}$ | $72.0_{\pm2.0}$ | $51.5_{\pm5.8}$ | $55.1_{\pm5.3}$ | $55.3_{\pm4.7}$ | $68.6_{\pm4.9}$ |
| | Negative | $34.5_{\pm4.4}$ | $44.6_{\pm2.9}$ | $51.6_{\pm1.6}$ | $57.8_{\pm1.3}$ | $34.5_{\pm3.4}$ | $39.7_{\pm3.7}$ | $42.7_{\pm3.2}$ | $41.5_{\pm2.3}$ |
| | Neutral | $34.2_{\pm20.7}$ | $47.9_{\pm3.6}$ | $52.7_{\pm3.9}$ | $53.0_{\pm2.0}$ | $40.3_{\pm4.9}$ | $45.5_{\pm4.3}$ | $45.0_{\pm3.3}$ | $53.0_{\pm2.9}$ |
| | Positive | $34.4_{\pm6.1}$ | $44.7_{\pm2.5}$ | $49.7_{\pm2.3}$ | $54.1_{\pm1.2}$ | $36.4_{\pm6.7}$ | $39.9_{\pm3.8}$ | $39.2_{\pm3.8}$ | $29.2_{\pm2.3}$ |
| | Very Positive | $59.6_{\pm11.8}$ | $61.7_{\pm3.5}$ | $70.3_{\pm3.2}$ | $70.3_{\pm1.3}$ | $50.0_{\pm6.4}$ | $53.6_{\pm5.3}$ | $54.1_{\pm4.5}$ | $42.0_{\pm2.1}$ |
| Rec.-b | Very Negative | $5.3_{\pm9.6}$ | $58.7_{\pm10.0}$ | $71.4_{\pm4.3}$ | $77.8_{\pm2.8}$ | $62.5_{\pm17.0}$ | $76.0_{\pm9.7}$ | $80.6_{\pm7.4}$ | $37.8_{\pm14.5}$ |
| | Negative | $75.2_{\pm16.7}$ | $48.5_{\pm10.8}$ | $55.3_{\pm4.7}$ | $47.9_{\pm5.5}$ | $19.1_{\pm17.7}$ | $24.6_{\pm12.9}$ | $22.5_{\pm10.5}$ | $55.7_{\pm10.9}$ |
| | Neutral | $2.5_{\pm3.3}$ | $40.4_{\pm10.1}$ | $46.8_{\pm10.2}$ | $59.7_{\pm4.4}$ | $39.2_{\pm18.4}$ | $35.0_{\pm11.1}$ | $35.6_{\pm11.2}$ | $14.1_{\pm5.1}$ |
| | Positive | $50.8_{\pm23.2}$ | $44.0_{\pm9.9}$ | $53.8_{\pm7.1}$ | $50.1_{\pm3.4}$ | $15.6_{\pm16.3}$ | $24.4_{\pm13.4}$ | $25.1_{\pm12.9}$ | $16.3_{\pm5.0}$ |
| | Very Positive | $56.2_{\pm25.0}$ | $70.2_{\pm8.7}$ | $64.4_{\pm7.5}$ | $72.2_{\pm2.1}$ | $86.6_{\pm7.4}$ | $83.7_{\pm6.7}$ | $83.3_{\pm6.6}$ | $94.0_{\pm1.8}$ |
| F1-b | Very Negative | $8.3_{\pm13.3}$ | $60.8_{\pm3.8}$ | $71.1_{\pm1.1}$ | $74.7_{\pm0.3}$ | $54.5_{\pm6.6}$ | $63.1_{\pm1.9}$ | $65.1_{\pm1.5}$ | $46.4_{\pm12.4}$ |
| | Negative | $46.2_{\pm3.7}$ | $46.0_{\pm6.5}$ | $53.2_{\pm1.6}$ | $52.1_{\pm2.7}$ | $20.4_{\pm14.7}$ | $28.5_{\pm10.6}$ | $28.1_{\pm9.4}$ | $46.9_{\pm2.8}$ |
| | Neutral | $4.3_{\pm5.5}$ | $42.9_{\pm5.7}$ | $48.6_{\pm4.3}$ | $56.0_{\pm1.0}$ | $37.1_{\pm11.3}$ | $38.2_{\pm7.1}$ | $38.6_{\pm7.5}$ | $21.7_{\pm6.1}$ |
| | Positive | $38.4_{\pm10.1}$ | $43.7_{\pm5.1}$ | $51.3_{\pm2.3}$ | $51.9_{\pm1.4}$ | $17.4_{\pm14.2}$ | $28.2_{\pm10.8}$ | $28.6_{\pm11.4}$ | $20.5_{\pm4.3}$ |
| | Very Positive | $52.4_{\pm11.8}$ | $65.2_{\pm2.5}$ | $66.8_{\pm2.9}$ | $71.2_{\pm0.4}$ | $62.7_{\pm3.4}$ | $64.9_{\pm2.5}$ | $65.2_{\pm1.6}$ | $58.0_{\pm1.8}$ |
| Prec.-l | Very Negative | $78.6_{\pm20.2}$ | $71.0_{\pm3.2}$ | $73.2_{\pm3.7}$ | $75.8_{\pm1.9}$ | $57.3_{\pm6.4}$ | $59.8_{\pm15.5}$ | $61.7_{\pm5.0}$ | $82.9_{\pm2.9}$ |
| | Negative | $39.4_{\pm5.2}$ | $53.0_{\pm1.9}$ | $54.9_{\pm1.5}$ | $60.9_{\pm1.3}$ | $42.9_{\pm6.8}$ | $45.2_{\pm11.2}$ | $50.3_{\pm2.9}$ | $44.7_{\pm2.6}$ |
| | Neutral | $39.2_{\pm22.8}$ | $54.8_{\pm3.0}$ | $57.7_{\pm2.7}$ | $58.1_{\pm1.9}$ | $48.0_{\pm5.9}$ | $46.7_{\pm11.8}$ | $52.1_{\pm2.6}$ | $64.2_{\pm2.8}$ |
| | Positive | $31.5_{\pm5.6}$ | $53.2_{\pm2.6}$ | $54.4_{\pm2.0}$ | $57.2_{\pm0.9}$ | $36.3_{\pm9.3}$ | $44.0_{\pm6.9}$ | $46.0_{\pm3.2}$ | $38.5_{\pm3.9}$ |
| | Very Positive | $72.1_{\pm8.9}$ | $70.7_{\pm6.1}$ | $72.7_{\pm3.1}$ | $73.1_{\pm2.5}$ | $52.6_{\pm8.4}$ | $58.1_{\pm15.1}$ | $60.7_{\pm4.1}$ | $56.2_{\pm5.3}$ |
| Rec.-l | Very Negative | $12.8_{\pm21.4}$ | $72.1_{\pm6.7}$ | $76.6_{\pm6.0}$ | $78.5_{\pm2.3}$ | $61.2_{\pm24.0}$ | $73.2_{\pm20.5}$ | $83.5_{\pm6.5}$ | $34.3_{\pm11.3}$ |
| | Negative | $55.0_{\pm23.9}$ | $47.0_{\pm6.8}$ | $56.8_{\pm6.2}$ | $53.1_{\pm5.3}$ | $29.0_{\pm18.3}$ | $26.5_{\pm14.9}$ | $31.2_{\pm9.6}$ | $75.8_{\pm5.1}$ |
| | Neutral | $4.4_{\pm7.3}$ | $60.8_{\pm7.6}$ | $53.8_{\pm6.9}$ | $62.6_{\pm3.1}$ | $40.0_{\pm14.5}$ | $42.0_{\pm17.1}$ | $40.2_{\pm8.8}$ | $20.7_{\pm8.5}$ |
| | Positive | $84.8_{\pm9.9}$ | $49.2_{\pm7.0}$ | $54.6_{\pm5.7}$ | $56.4_{\pm2.1}$ | $22.3_{\pm22.2}$ | $41.1_{\pm19.9}$ | $37.4_{\pm12.3}$ | $36.8_{\pm13.4}$ |
| | Very Positive | $36.7_{\pm22.8}$ | $71.9_{\pm12.4}$ | $69.7_{\pm7.1}$ | $74.5_{\pm5.1}$ | $88.3_{\pm10.2}$ | $76.6_{\pm21.8}$ | $85.6_{\pm5.1}$ | $88.8_{\pm8.6}$ |
| F1-l | Very Negative | $16.2_{\pm20.5}$ | $71.2_{\pm2.3}$ | $74.6_{\pm1.2}$ | $77.1_{\pm0.4}$ | $55.7_{\pm11.8}$ | $64.5_{\pm15.3}$ | $70.5_{\pm1.7}$ | $47.2_{\pm11.9}$ |
| | Negative | $43.2_{\pm12.3}$ | $49.5_{\pm3.4}$ | $55.6_{\pm2.6}$ | $56.6_{\pm2.6}$ | $30.1_{\pm13.6}$ | $31.1_{\pm13.6}$ | $37.7_{\pm7.6}$ | $56.0_{\pm1.1}$ |
| | Neutral | $7.0_{\pm10.5}$ | $57.2_{\pm2.4}$ | $55.3_{\pm3.0}$ | $60.2_{\pm0.9}$ | $42.0_{\pm9.6}$ | $42.4_{\pm12.3}$ | $44.7_{\pm5.2}$ | $30.2_{\pm9.9}$ |
| | Positive | $45.4_{\pm5.6}$ | $50.8_{\pm3.1}$ | $54.3_{\pm2.1}$ | $56.8_{\pm1.0}$ | $22.5_{\pm17.7}$ | $39.4_{\pm9.3}$ | $40.1_{\pm8.8}$ | $36.8_{\pm7.7}$ |
| | Very Positive | $44.2_{\pm22.2}$ | $70.1_{\pm4.1}$ | $70.8_{\pm2.5}$ | $73.6_{\pm1.4}$ | $64.8_{\pm4.0}$ | $64.6_{\pm15.9}$ | $70.8_{\pm1.4}$ | $68.2_{\pm2.4}$ |

Table 26: Precision, recall, and F1 for Yelp-5, where 'b' refers to RoBERTa-base, 'l' refers to RoBERTa-large.

| Yelp-2 | | zero-shot | In-domain | | | Out-of-domain | | | LDT |
|---|---|---|---|---|---|---|---|---|---|
| | | | 10 | 100 | 500 | 10 | 100 | 500 | |
| Prec.-b | Negative | $91.8_{\pm26.4}$ | $92.6_{\pm2.3}$ | $94.1_{\pm1.9}$ | $95.3_{\pm1.8}$ | $94.8_{\pm2.0}$ | $94.3_{\pm2.5}$ | $92.8_{\pm2.5}$ | $97.0_{\pm0.9}$ |
| | Positive | $58.8_{\pm7.3}$ | $93.5_{\pm4.3}$ | $95.3_{\pm1.7}$ | $95.1_{\pm2.7}$ | $89.6_{\pm3.2}$ | $88.9_{\pm3.7}$ | $92.5_{\pm3.6}$ | $82.5_{\pm3.8}$ |
| Rec.-b | Negative | $27.6_{\pm21.7}$ | $93.3_{\pm5.1}$ | $95.3_{\pm1.9}$ | $94.9_{\pm3.3}$ | $88.8_{\pm4.1}$ | $87.9_{\pm4.7}$ | $92.2_{\pm4.6}$ | $79.1_{\pm5.7}$ |
| | Positive | $99.7_{\pm0.3}$ | $92.4_{\pm2.8}$ | $94.0_{\pm2.2}$ | $95.3_{\pm2.0}$ | $95.1_{\pm2.2}$ | $94.5_{\pm2.9}$ | $92.7_{\pm2.9}$ | $97.5_{\pm0.9}$ |
| F1-b | Negative | $38.7_{\pm27.9}$ | $92.8_{\pm2.0}$ | $94.7_{\pm0.6}$ | $95.1_{\pm1.4}$ | $91.6_{\pm1.9}$ | $90.9_{\pm1.9}$ | $92.4_{\pm1.8}$ | $87.0_{\pm3.2}$ |
| | Positive | $73.7_{\pm5.7}$ | $92.8_{\pm1.5}$ | $94.6_{\pm0.7}$ | $95.1_{\pm1.0}$ | $92.2_{\pm1.4}$ | $91.5_{\pm1.4}$ | $92.5_{\pm1.2}$ | $89.3_{\pm1.9}$ |
| Prec.-l | Negative | $99.5_{\pm0.4}$ | $95.7_{\pm1.8}$ | $95.7_{\pm1.7}$ | $97.1_{\pm0.8}$ | $96.6_{\pm1.0}$ | $95.5_{\pm1.9}$ | $95.8_{\pm1.7}$ | $97.6_{\pm0.9}$ |
| | Positive | $65.5_{\pm13.5}$ | $95.9_{\pm2.9}$ | $97.3_{\pm0.9}$ | $97.2_{\pm0.6}$ | $95.0_{\pm1.6}$ | $94.5_{\pm2.5}$ | $95.3_{\pm2.0}$ | $92.2_{\pm3.3}$ |
| Rec.-l | Negative | $41.4_{\pm31.7}$ | $95.7_{\pm3.2}$ | $97.4_{\pm1.0}$ | $97.2_{\pm0.7}$ | $94.9_{\pm1.8}$ | $94.3_{\pm3.0}$ | $95.2_{\pm2.2}$ | $91.6_{\pm4.1}$ |
| | Positive | $99.7_{\pm0.3}$ | $95.7_{\pm1.9}$ | $95.6_{\pm1.9}$ | $97.1_{\pm0.8}$ | $96.7_{\pm1.1}$ | $95.5_{\pm2.1}$ | $95.7_{\pm1.9}$ | $97.7_{\pm0.9}$ |
| F1-l | Negative | $51.5_{\pm33.7}$ | $95.7_{\pm1.0}$ | $96.5_{\pm0.6}$ | $97.1_{\pm0.2}$ | $95.7_{\pm0.7}$ | $94.8_{\pm1.2}$ | $95.4_{\pm0.8}$ | $94.4_{\pm2.1}$ |
| | Positive | $78.3_{\pm9.6}$ | $95.7_{\pm0.8}$ | $96.4_{\pm0.8}$ | $97.1_{\pm0.3}$ | $95.8_{\pm0.6}$ | $94.9_{\pm1.0}$ | $95.5_{\pm0.7}$ | $94.8_{\pm1.5}$ |

Table 27: Precision, recall, and F1 for Yelp-2, where 'b' refers to RoBERTa-base, 'l' refers to RoBERTa-large.

| SST-5 | | zero-shot | In-domain | | | Out-of-domain | | | LDT |
|---|---|---|---|---|---|---|---|---|---|
| | | | 10 | 100 | 500 | 10 | 100 | 500 | |
| Prec.-b | Very Negative | $13.7_{\pm15.1}$ | $34.9_{\pm4.3}$ | $43.4_{\pm5.3}$ | $44.7_{\pm3.9}$ | $28.5_{\pm4.0}$ | $31.8_{\pm3.7}$ | $33.0_{\pm2.7}$ | $26.9_{\pm3.1}$ |
| | Negative | $45.0_{\pm7.7}$ | $52.2_{\pm2.6}$ | $55.6_{\pm1.9}$ | $59.0_{\pm1.9}$ | $53.0_{\pm2.8}$ | $53.5_{\pm2.1}$ | $58.6_{\pm2.7}$ | $53.2_{\pm1.6}$ |
| | Neutral | $8.9_{\pm11.5}$ | $25.8_{\pm3.5}$ | $32.2_{\pm2.9}$ | $34.8_{\pm2.1}$ | $25.9_{\pm6.2}$ | $27.7_{\pm4.5}$ | $26.0_{\pm1.8}$ | $26.0_{\pm3.3}$ |
| | Positive | $33.0_{\pm5.4}$ | $44.5_{\pm4.6}$ | $49.1_{\pm2.2}$ | $51.8_{\pm2.7}$ | $41.6_{\pm4.7}$ | $42.8_{\pm4.4}$ | $43.5_{\pm5.5}$ | $39.5_{\pm2.4}$ |
| | Very Positive | $45.9_{\pm21.1}$ | $54.2_{\pm6.3}$ | $58.1_{\pm4.7}$ | $57.9_{\pm3.2}$ | $47.7_{\pm6.3}$ | $49.9_{\pm5.2}$ | $49.5_{\pm2.4}$ | $42.5_{\pm3.3}$ |
| Rec.-b | Very Negative | $4.4_{\pm8.1}$ | $37.1_{\pm10.8}$ | $53.2_{\pm11.0}$ | $57.6_{\pm8.8}$ | $48.9_{\pm22.3}$ | $36.0_{\pm9.4}$ | $64.0_{\pm6.6}$ | $24.2_{\pm7.4}$ |
| | Negative | $60.3_{\pm29.8}$ | $47.4_{\pm9.1}$ | $44.9_{\pm11.8}$ | $42.9_{\pm7.0}$ | $45.7_{\pm23.9}$ | $59.6_{\pm10.5}$ | $38.5_{\pm6.8}$ | $58.0_{\pm7.4}$ |
| | Neutral | $0.7_{\pm1.8}$ | $24.2_{\pm12.1}$ | $32.8_{\pm9.5}$ | $34.5_{\pm8.5}$ | $13.2_{\pm12.7}$ | $23.1_{\pm16.1}$ | $37.7_{\pm5.8}$ | $11.7_{\pm7.6}$ |
| | Positive | $57.5_{\pm27.5}$ | $46.6_{\pm9.5}$ | $51.4_{\pm8.9}$ | $53.2_{\pm8.4}$ | $30.7_{\pm20.6}$ | $32.5_{\pm18.0}$ | $14.8_{\pm5.0}$ | $30.1_{\pm9.4}$ |
| | Very Positive | $24.3_{\pm27.2}$ | $56.0_{\pm11.4}$ | $57.5_{\pm9.4}$ | $66.9_{\pm7.3}$ | $65.6_{\pm13.9}$ | $52.2_{\pm14.5}$ | $62.1_{\pm7.0}$ | $73.6_{\pm7.0}$ |
| F1-b | Very Negative | $5.7_{\pm8.6}$ | $35.0_{\pm5.1}$ | $46.6_{\pm3.2}$ | $49.7_{\pm2.4}$ | $33.2_{\pm5.4}$ | $32.8_{\pm3.1}$ | $43.2_{\pm1.1}$ | $24.7_{\pm4.3}$ |
| | Negative | $46.4_{\pm8.6}$ | $49.1_{\pm4.8}$ | $48.7_{\pm6.8}$ | $49.3_{\pm4.2}$ | $45.1_{\pm17.1}$ | $55.8_{\pm4.4}$ | $46.0_{\pm4.9}$ | $55.2_{\pm3.0}$ |
| | Neutral | $1.2_{\pm2.9}$ | $23.5_{\pm8.2}$ | $31.5_{\pm4.6}$ | $34.0_{\pm4.0}$ | $13.9_{\pm9.8}$ | $21.4_{\pm7.1}$ | $30.6_{\pm2.4}$ | $14.9_{\pm6.6}$ |
| | Positive | $38.1_{\pm11.4}$ | $44.8_{\pm4.6}$ | $49.8_{\pm4.0}$ | $52.0_{\pm4.0}$ | $31.7_{\pm15.2}$ | $34.2_{\pm13.0}$ | $21.7_{\pm5.7}$ | $33.3_{\pm6.8}$ |
| | Very Positive | $22.9_{\pm17.2}$ | $53.9_{\pm3.4}$ | $57.1_{\pm4.3}$ | $61.7_{\pm2.0}$ | $53.8_{\pm3.5}$ | $49.4_{\pm5.5}$ | $54.8_{\pm2.2}$ | $53.6_{\pm1.5}$ |
| Prec.-l | Very Negative | $21.1_{\pm20.5}$ | $35.2_{\pm4.8}$ | $41.1_{\pm10.7}$ | $45.7_{\pm3.0}$ | $34.1_{\pm5.6}$ | $33.0_{\pm3.7}$ | $33.2_{\pm3.5}$ | $35.9_{\pm6.0}$ |
| | Negative | $51.7_{\pm4.7}$ | $52.0_{\pm3.4}$ | $55.0_{\pm13.0}$ | $59.6_{\pm1.8}$ | $53.5_{\pm4.1}$ | $53.2_{\pm3.4}$ | $58.1_{\pm2.3}$ | $54.5_{\pm1.4}$ |
| | Neutral | $7.5_{\pm13.0}$ | $26.6_{\pm2.7}$ | $33.4_{\pm8.2}$ | $38.6_{\pm2.5}$ | $33.3_{\pm3.0}$ | $29.1_{\pm4.5}$ | $31.6_{\pm2.8}$ | $32.3_{\pm6.8}$ |
| | Positive | $31.3_{\pm6.1}$ | $46.6_{\pm7.2}$ | $49.1_{\pm6.7}$ | $55.8_{\pm1.9}$ | $48.1_{\pm4.0}$ | $49.0_{\pm4.6}$ | $51.1_{\pm4.1}$ | $46.2_{\pm2.6}$ |
| | Very Positive | $66.8_{\pm25.6}$ | $53.2_{\pm5.8}$ | $56.6_{\pm13.8}$ | $61.8_{\pm3.1}$ | $57.1_{\pm7.6}$ | $52.9_{\pm4.4}$ | $50.1_{\pm4.6}$ | $53.4_{\pm6.4}$ |
| Rec.-l | Very Negative | $13.6_{\pm23.6}$ | $39.1_{\pm9.9}$ | $54.5_{\pm16.7}$ | $58.1_{\pm6.2}$ | $56.7_{\pm22.3}$ | $58.3_{\pm11.9}$ | $69.6_{\pm7.8}$ | $21.2_{\pm7.4}$ |
| | Negative | $36.7_{\pm26.8}$ | $43.1_{\pm12.5}$ | $37.4_{\pm15.2}$ | $45.9_{\pm7.2}$ | $48.1_{\pm20.3}$ | $54.2_{\pm10.7}$ | $41.4_{\pm8.3}$ | $73.7_{\pm6.5}$ |
| | Neutral | $0.7_{\pm1.7}$ | $25.3_{\pm10.5}$ | $33.0_{\pm15.4}$ | $38.2_{\pm7.6}$ | $18.3_{\pm11.8}$ | $16.6_{\pm10.0}$ | $25.8_{\pm5.6}$ | $6.7_{\pm3.8}$ |
| | Positive | $91.5_{\pm9.5}$ | $43.2_{\pm10.8}$ | $58.1_{\pm14.7}$ | $55.2_{\pm7.7}$ | $43.0_{\pm13.1}$ | $30.8_{\pm9.7}$ | $28.8_{\pm8.0}$ | $52.6_{\pm12.7}$ |
| | Very Positive | $8.3_{\pm12.0}$ | $65.4_{\pm10.2}$ | $58.4_{\pm19.2}$ | $72.2_{\pm6.8}$ | $59.8_{\pm14.3}$ | $63.5_{\pm10.2}$ | $68.0_{\pm11.1}$ | $67.2_{\pm13.5}$ |
| F1-l | Very Negative | $11.2_{\pm14.9}$ | $36.3_{\pm5.5}$ | $45.7_{\pm10.9}$ | $50.9_{\pm1.7}$ | $39.7_{\pm3.9}$ | $41.2_{\pm2.6}$ | $44.6_{\pm2.6}$ | $25.8_{\pm5.7}$ |
| | Negative | $37.6_{\pm21.2}$ | $45.9_{\pm9.0}$ | $43.1_{\pm13.9}$ | $51.5_{\pm4.7}$ | $47.7_{\pm11.8}$ | $52.9_{\pm4.6}$ | $47.7_{\pm5.3}$ | $62.5_{\pm2.0}$ |
| | Neutral | $1.2_{\pm2.9}$ | $24.8_{\pm5.7}$ | $31.4_{\pm9.8}$ | $37.9_{\pm3.3}$ | $21.3_{\pm8.7}$ | $19.0_{\pm6.6}$ | $28.0_{\pm3.4}$ | $10.8_{\pm5.5}$ |
| | Positive | $46.0_{\pm5.8}$ | $43.3_{\pm8.6}$ | $51.5_{\pm5.5}$ | $55.1_{\pm3.5}$ | $44.0_{\pm6.2}$ | $37.0_{\pm7.2}$ | $36.1_{\pm6.9}$ | $48.2_{\pm4.9}$ |
| | Very Positive | $12.1_{\pm15.0}$ | $57.7_{\pm2.0}$ | $55.9_{\pm14.6}$ | $66.3_{\pm1.7}$ | $56.5_{\pm5.8}$ | $57.0_{\pm3.1}$ | $56.8_{\pm2.7}$ | $58.0_{\pm4.0}$ |

Table 28: Precision, recall, and F1 for SST-5, where 'b' refers to RoBERTa-base, 'l' refers to RoBERTa-large.

| SST-2 | | zero-shot | In-domain | | | Out-of-domain | | | LDT |
|---|---|---|---|---|---|---|---|---|---|
| | | | 10 | 100 | 500 | 10 | 100 | 500 | |
| Prec.-b | Negative | $59.6_{\pm11.8}$ | $91.1_{\pm3.0}$ | $93.7_{\pm2.3}$ | $93.3_{\pm2.6}$ | $74.1_{\pm8.3}$ | $79.9_{\pm8.0}$ | $83.6_{\pm5.9}$ | $81.8_{\pm4.0}$ |
| | Positive | $58.8_{\pm8.4}$ | $88.0_{\pm4.0}$ | $88.6_{\pm4.5}$ | $91.9_{\pm3.0}$ | $95.3_{\pm4.0}$ | $94.1_{\pm3.4}$ | $91.9_{\pm5.2}$ | $88.3_{\pm3.1}$ |
| Rec.-b | Negative | $28.9_{\pm25.3}$ | $87.2_{\pm5.5}$ | $87.5_{\pm6.3}$ | $91.7_{\pm3.7}$ | $96.1_{\pm4.5}$ | $94.8_{\pm3.9}$ | $92.2_{\pm6.8}$ | $89.1_{\pm3.9}$ |
| | Positive | $96.5_{\pm3.6}$ | $91.2_{\pm3.6}$ | $93.9_{\pm2.7}$ | $93.3_{\pm3.0}$ | $64.1_{\pm16.4}$ | $74.4_{\pm14.6}$ | $80.8_{\pm9.2}$ | $79.9_{\pm5.9}$ |
| F1-b | Negative | $37.7_{\pm29.6}$ | $88.9_{\pm2.3}$ | $90.3_{\pm3.0}$ | $92.4_{\pm1.2}$ | $83.2_{\pm4.3}$ | $86.3_{\pm3.8}$ | $87.2_{\pm2.7}$ | $85.2_{\pm1.9}$ |
| | Positive | $72.6_{\pm5.4}$ | $89.4_{\pm1.5}$ | $91.0_{\pm1.8}$ | $92.5_{\pm0.9}$ | $75.1_{\pm11.8}$ | $81.9_{\pm10.4}$ | $85.4_{\pm4.2}$ | $83.7_{\pm2.9}$ |
| Prec.-l | Negative | $83.2_{\pm35.3}$ | $94.5_{\pm2.3}$ | $94.8_{\pm2.2}$ | $95.1_{\pm1.7}$ | $88.3_{\pm3.9}$ | $84.9_{\pm6.2}$ | $88.7_{\pm4.1}$ | $94.5_{\pm2.3}$ |
| | Positive | $59.9_{\pm11.2}$ | $92.1_{\pm3.2}$ | $91.9_{\pm4.0}$ | $93.9_{\pm2.1}$ | $94.2_{\pm3.5}$ | $95.8_{\pm3.2}$ | $94.5_{\pm2.5}$ | $89.0_{\pm3.9}$ |
| Rec.-l | Negative | $28.8_{\pm29.9}$ | $91.7_{\pm3.9}$ | $91.4_{\pm5.5}$ | $93.8_{\pm2.4}$ | $94.4_{\pm4.0}$ | $96.0_{\pm3.8}$ | $94.7_{\pm2.8}$ | $88.0_{\pm5.4}$ |
| | Positive | $98.8_{\pm1.6}$ | $94.5_{\pm2.7}$ | $94.8_{\pm2.4}$ | $95.1_{\pm1.9}$ | $87.0_{\pm5.2}$ | $82.0_{\pm9.3}$ | $87.5_{\pm5.6}$ | $94.7_{\pm2.5}$ |
| F1-l | Negative | $36.5_{\pm35.1}$ | $93.0_{\pm1.3}$ | $92.9_{\pm2.7}$ | $94.4_{\pm0.7}$ | $91.1_{\pm1.4}$ | $89.9_{\pm2.8}$ | $91.5_{\pm1.5}$ | $91.0_{\pm2.6}$ |
| | Positive | $74.0_{\pm7.9}$ | $93.2_{\pm0.9}$ | $93.2_{\pm1.7}$ | $94.4_{\pm0.5}$ | $90.3_{\pm1.8}$ | $87.9_{\pm4.9}$ | $90.7_{\pm2.4}$ | $91.7_{\pm1.7}$ |

Table 29: Precision, recall, and F1 for SST-2, where 'b' refers to RoBERTa-base, 'l' refers to RoBERTa-large.