# OpenReview forum: "The Benefits of Label-Description Training for Zero-Shot Text Classification"
_EMNLP/2023/Conference — EMNLP 2023 Main_

### Official Review · Reviewer_ZfZP · 2023-07-19

**Soundness:** 4

**Excitement:**

4: Strong: This paper deepens the understanding of some phenomenon or lowers the barriers to an existing research direction.

**Paper Topic And Main Contributions:**

The paper proposes a new paradigm for performing zero-shot text classification in the following way:
* Several types of label descriptions are used create a more detailed description of each class label, including keywords, short sentences/templates, and Wikipedia definitions.
* A pretrained text encoder model (e.g., RoBERTa) is fine-tuned on such small label description datasets.
* The fine-tuned model is taken for inference on task-specific data in prompting format.

The authors evaluate on several topic and sentiment classification tasks and demonstrate that the proposed paradigm is able to outperform conventional zero-shot inference by large margins.

**Questions For The Authors:**

Please clarify how the fine-tuning on LabelDesc data is exactly conducted. I'd consider updating my rating given the authors' response of this.

**Reasons To Accept:**

* Novelty: The new paradigm for zero-shot text classification is interesting and novel based on my knowledge. This is a new attempt aiming to leverage generic label descriptions to boost zero-shot performance.
* Comprehensive evaluation: The experiments are conducted on both topic and sentiment classification tasks which seem sufficient to me. There are also analytical studies regarding the sensitivity to prompt patterns/verbalizers and domain transfer ability.
* Effectiveness: The new method is empirically effective across standard text classification benchmarks, outperforming previous standard zero-shot prompting and achieving similar performance to larger GPT3 models with RoBERTa-sized encoders.

**Reasons To Reject:**

* Presentation of the method: While the motivation and intuition is general clear with the label description training, I didn't find a very clear and concrete introduction of how the fine-tuning is exactly conducted (starting Line 271). Specifically, how would you train the model on class-representative keywords, short templates, and Wikipedia descriptions, respectively? What is the input format and the prediction target? This might be straightforward, but I'd appreciate a figure or at least some detailed descriptions with regard to this, which will help the readers understand the method better.

**Reproducibility:**

5: Could easily reproduce the results.

**Reviewer Confidence:**

5: Positive that my evaluation is correct. I read the paper very carefully and I am very familiar with related work.

---

> ### Author Rebuttal · Authors · 2023-08-29
>
> Thank you for your comments! We’ll be sure to add more details (descriptions and figures) to the camera ready. Below we describe exactly how the training data is formatted during the training.
>
> For topic classification, let's say, we are using the label “sports” (used both in AGNews and Yahoo), whose verbalizer is also chosen as “sports”. We illustrate a training example here. With the related term “racing” (see Table 1 in the paper) we create an input as follows: “racing Question: What is the topic of this article? Answer: [MASK].”, and the model will be trained to output the correct verbalizer “sports” using the masked language modeling head at the position of [MASK]. The related term “racing” could be replaced by the dictionary definition, other synonyms, or Wikipedia sentences. For instance, Table 1 presents the dictionary definition.
>
> For topic classification in the inference time, assume that the original text input is “Need for carbon sink technologies Climate scientists tell a conference that greater efforts should be made to pull CO2 from the atmosphere.” from AGNews, the input to the model is as follows using pattern 1: “Need for carbon sink technologies Climate scientists tell a conference that greater efforts should be made to pull CO2 from the atmosphere. Question: What is the topic of this article? Answer: [MASK].”, and the model is expected to output the correct verbalizer “Tech” (gold label is “Sci/Tech”).
>
> For sentiment classification, let's say we are curating an example for the label “Very Negative” (SST-5 or yelp-5 dataset), whose verbalizer is chosen as “terrible” for illustrations of short templates. We choose “awful” as a synonym term of the verbalizer, taking the template “It was t.” as an example, the input is as follows: “It was awful. Question: What is the sentiment of this text? Answer: [MASK].”, and the model will be trained to output the correct verbalizer “terrible”.
>
> During inference, we follow the same procedures as in topic classification. Assume that the original text input is “Effective but too-tepid biopic” from SST-5, with pattern 1 for SST-5, the input to the model is as follows “Effective but too-tepid biopic Question: What is the sentiment of this text? Answer: [MASK].”, and the model is expected to output the correct verbalizer “okay”, as the gold label is “Neutral” for this text example.

---

### Official Review · Reviewer_S9DN · 2023-07-25

**Soundness:** 4

**Excitement:**

4: Strong: This paper deepens the understanding of some phenomenon or lowers the barriers to an existing research direction.

**Paper Topic And Main Contributions:**

This paper proposes a new way to construct a dataset which could obviously improve the performance of LLMs on zero-shot text classification tasks. The data in the dataset constructed by this method only contains the description of the labels. In the topic classification and sentiment classification tasks in this paper, the model using this method outperform than the other baseline models with zero-shot.

**Reasons To Accept:**

This paper proposes a new way of constructing dataset which only consists of the description of the labels for LLMs on zero-shot text classification tasks. And this method is not strongly dependent on patterns and verbalizers. It is also hopefully to applied on other tasks.

**Reasons To Reject:**

Though the method proposed by this paper could obviously improve the performance of LLMs on zero-shot text classification tasks, it's a simple method which has a strong rely on the practitioner’s understanding of the labels and usage situation. It also has a need of a pre-trained model which could produce some potential biases.

**Reproducibility:**

5: Could easily reproduce the results.

**Reviewer Confidence:**

4: Quite sure. I tried to check the important points carefully. It's unlikely, though conceivable, that I missed something that should affect my ratings.

---

> ### Author Rebuttal · Authors · 2023-08-29
>
> Thank you for the review!
>
> Regarding practitioners’ understanding of the labels: A significant aim of this project is to demonstrate how simply curating a few training instances for each label can substantially enhance performance over zero-shot methods. Even though this does require some understanding from the practitioners, it's typically a simple task involving the gathering of synonyms or online dictionary definition. Furthermore, it only takes a few minutes to collect these instances. We've demonstrated that just a couple of such examples are adequate to create a more efficient classifier than a zero-shot one across all the data sets and domains we used.
> Regarding potential bias - although we mentioned this in the limitation section (line 616) our examples are largely very objective (e.g., a dictionary definition) which limits the tendency to result in additional bias.

---

### Official Review · Reviewer_RAHH · 2023-08-04

**Soundness:** 3

**Excitement:**

2: Mediocre: This paper makes marginal contributions (vs non-contemporaneous work), so I would rather not see it in the conference.

**Paper Topic And Main Contributions:**

This paper proposes a label-description training approach to improve the classification performance of pre-trained MLMs when using a pattern-verbalizer approach. Depending on the classification task, the authors create a dataset where the labels are the labels of the task and the inputs are based on descriptions, relevant topics, and/or simple patterns. They then fine-tune an MLM after converting the data using a pattern-verbalizer approach.

Their results show that fine-tuning the MLM on the created dataset is better and more robust than using the MLM out-of-the-box. The authors demonstrate the robustness of their approach with ablation tests where they use random verbalizers, shuffled verbalizers, and no verbalizers (and patterns) at all. Finally, they compare their approach with in-domain and out-of-domain few-shot learning approaches.

**Questions For The Authors:**

* Line 485: The authors mention that the models were trained for 15 epochs. Do the authors use the checkpoint from the best performing epoch or do they use the last checkpoint? If the latter why?

**Reasons To Accept:**

* The paper is well-written and easy to understand.
* The approach is straightforward and seems to improve over the pattern-verbalizer approach.
* The idea to construct synthetic data using only the labels is interesting.
* The authors have included ablation tests to demonstrate the robustness of their approach.

**Reasons To Reject:**

* The results are neither surprising nor exciting. Zero-shot and few-shot learning with GPT-3 using prompting had still a large gap with supervised SotA models. On the other hand, InstructGPT [1] was much better than GPT-3 and the main reason was that it was trained to follow instructions (prompts). The proposed approach is very similar to the basic idea of InstructGPT. The authors fine-tune an MLM using patterns, which can be seen as instructions, and synthetic data. In effect, the model has learned to use these patterns and is therefore able to use this knowledge for classification.
* The randomized and mismatched ablations are not very well designed. The authors use the same setting found during hyper-parameter tuning. However, the ablated versions have a more difficult task to solve. The randomized version has to learn new embeddings from scratch and the mismatched version has to learn a new meaning for each label. Thus the model may require more steps to adapt. I would be interested to see how the ablated versions would perform if the authors tuned the hyper-parameters on 20NG for each of these settings.
* I believe the authors should have included one more baseline, i.e., the model trained on the 20NG dataset.
* I am not entirely convinced by the choice of the authors to use only 4 labels from the 20NG dataset. For instance, similar labels could have been merged. In addition, the proposed approach could include more labels, e.g., from Wikipedia, which may or may not contain the labels of the downstream tasks, and create a larger dataset. This combined with the baseline I proposed above could lead to a general-purpose pattern-based classifier.
* The results with InstructGPT (text-davinci-003) cannot be compared with the other results. It is understandable that the authors were able to only evaluate 1,000 samples to reduce the cost. I would expect however to compare these results with the other methods on the same set of samples.

[1] https://arxiv.org/abs/2203.02155

**Reproducibility:**

4: Could mostly reproduce the results, but there may be some variation because of sample variance or minor variations in their interpretation of the protocol or method.

**Reviewer Confidence:**

5: Positive that my evaluation is correct. I read the paper very carefully and I am very familiar with related work.

**Typos Grammar Style And Presentation Improvements:**

* Lines 285 - 287: This part is a bit unclear. The authors mention both steps and epochs. I would them to evaluate every $n$ step or define the number of steps per epoch.
* The Abstract and the Introduction begin by stating that LLMs have improved zero-shot text classification. Thus, they give the impression that the authors will improve the zero-shot capabilities of LLMs, but the authors focus on RoBERTa which cannot be considered an LLM.

---

> ### Author Rebuttal · Authors · 2023-08-29
>
> Thanks for your comments and review!
>
> Regarding the InstructGPT comparison: InstructGPT incorporates reinforcement learning from human feedback (RLHF) to finetune the same model to follow a broad class of written instructions, thus enabling the model for cross-task generalization. However, we only use one written pattern for finetuning the model without human feedback (our different patterns correspond to different finetuned models), keeping the same pattern for training and testing instead of different instructions, and we use the same set of patterns for comparison between zero-shot and LabelDescTraining. Therefore, our approach differs significantly from InstructGPT.
>
> Regarding the hyperparameters for the ablations, we actually tuned hyperparameters (on 20NG) independently for each ablation setting. Please refer to Table 13 in Appendix A.3, where we explain our tuned parameters. We’ll be sure to add more details to the next version.
>
> Regarding a baseline trained on 20NG, thanks for your suggestion! We can easily add results of this model to the next version.
>
> Regarding choosing only 4 labels from 20NG for tuning, we agree that many other choices could be made here that could potentially improve our results overall. The four classes we chose have a similar topical granularity as the classes in widely used datasets like AGNews and Yahoo. If we had grouped labels or used more labels, we could potentially have found even better hyperparameters, but we did not consider different variations here. Thanks for the suggestion regarding moving in the direction of building general purpose classifiers. Some of the cited work we compare to, like Chu et al., does this. However, our paper instead starts with a given label set and seeks to build a classifier that is independent of the input text domain for that label set by using label descriptions.
>
> Regarding the results with text-davinci-003: We want to mention here that we did compare extensively to available SOTA results (Table 4) for all datasets. The reason we reported text-davinci-003 results is to show that our method not only works for MLM-style models like RoBERTa, but also works for autoregressive models like text-davinci-003. Although we have used limited data for text-davinci-003 mostly for cost reasons, the results present a similar pattern (label description training is better for topic datasets and highly comparable to other methods even against text-davinci-003).
>
> Question 1: We use the checkpoint from the best performing epoch selected by the dev set.
>
> Thanks for your suggestions regarding the writing. We are evaluating every 24 steps (24 is the LabelDesc dataset size for 20NG, namely the number of steps per epoch).
>
> Regarding the use of the term LLM: we used the term LLM to represent any pretrained language model, and we’ll clarify this.

---

### Meta-Review · Area_Chair_SQ5w · 2023-09-19

**Recommendation:** 5

**Metareview:**

The paper creates a dataset of label descriptions which can help models improve zero-shot classification.

The idea is interesting and experiments sufficiently demonstrate the usefulness.
The reviewers feel that most of the paper is clear and easy to follow.

The baselines and comparison to existing methods could be further improved, following specific suggestions from the reviewers.
Certain parts of the method could be clarified, as pointed out by reviewer ZfZP.

---

### Decision · Program_Chairs · 2023-10-07

**Decision:**

Accept-Main

**Comment:**

The paper creates a dataset of label descriptions which can help models improve zero-shot classification.

The idea is interesting and experiments sufficiently demonstrate the usefulness.
The reviewers feel that most of the paper is clear and easy to follow.

The baselines and comparison to existing methods could be further improved, following specific suggestions from the reviewers.
Certain parts of the method could be clarified, as pointed out by reviewer ZfZP.